# Loss-of-function mutation in PRMT9 causes abnormal synapse development by dysregulation of RNA alternative splicing

Lei Shen[1,10], Xiaokuang Ma [2,10], Yuanyuan Wang[3,4,10], Zhihao Wang[1], Yi Zhang[1], Hoang Quoc Hai Pham [1,5], Xiaoqun Tao[1,5], Yuehua Cui[2], Jing Wei [2], Dimitri Lin[1], Tharindumala Abeywanada[1], Swanand Hardikar[6], Levon Halabelian [7], Noah Smith[4], Taiping Chen [6], Dalia Barsyte-Lovejoy [7], Shenfeng Qiu [2] ✉, Yi Xing [4,8,9] ✉ & Yanzhong Yang [1,5] ✉

Protein arginine methyltransferase 9 (PRMT9) is a recently identified member of the PRMT family, yet its biological function remains largely unknown. Here, by characterizing an intellectual disability associated PRMT9 mutation (G189R) and establishing a *Prmt9* conditional knockout (cKO) mouse model, we uncover an important function of PRMT9 in neuronal development. The G189R mutation abolishes PRMT9 methyltransferase activity and reduces its protein stability. Knockout of *Prmt9* in hippocampal neurons causes alternative splicing of ~1900 genes, which likely accounts for the aberrant synapse development and impaired learning and memory in the *Prmt9* cKO mice. Mechanistically, we discover a methylation-sensitive protein–RNA interaction between the arginine 508 (R508) of the splicing factor 3B subunit 2 (SF3B2), the site that is exclusively methylated by PRMT9, and the pre-mRNA anchoring site, a cis-regulatory element that is critical for RNA splicing. Additionally, using human and mouse cell lines, as well as an SF3B2 arginine methylation-deficient mouse model, we provide strong evidence that SF3B2 is the primary methylation substrate of PRMT9, thus highlighting the conserved function of the PRMT9/SF3B2 axis in regulating pre-mRNA splicing.

Arginine methylation is a widespread post-translational modification (PTM) found in all eukaryotes[1,2]. It is catalyzed by a family of enzymes known as protein arginine methyltransferases (PRMTs). The human PRMT family consists of nine members (PRMT1–9) that exhibit different substrate specificities and generate distinct methylation states, including monomethylation (MMA), asymmetric dimethylation (ADMA), and symmetric dimethylation (SDMA). These PTMs are involved in a wide range of cellular functions, including gene expression, RNA processing, genome stability, and signal transduction[1,2]. Aberrant protein arginine methylation underlies many human

[1]Department of Cancer Genetics and Epigenetics, Beckman Research Institute, City of Hope Cancer Center, Duarte, CA 91010, USA. [2]Basic Medical Sciences, University of Arizona College of Medicine-Phoenix, Phoenix, AZ 85004, USA. [3]Bioinformatics Interdepartmental Graduate Program, University of California, Los Angeles, CA 90095, USA. [4]Center for Computational and Genomic Medicine, The Children's Hospital of Philadelphia, Philadelphia, PA 19104, USA. [5]Irell & Manella Graduate School of Biological Sciences, Beckman Research Institute of City of Hope, Duarte, CA 91010, USA. [6]Department of Epigenetics and Molecular Carcinogenesis, The University of Texas MD Anderson Cancer Center, Houston, TX 77030, USA. [7]Structural Genomics Consortium, University of Toronto, Toronto, ON, Canada. [8]Department of Pathology and Laboratory Medicine, University of Pennsylvania, Philadelphia, PA 19104, USA. [9]Department of Biomedical and Health Informatics, The Children's Hospital of Philadelphia, Philadelphia, PA 19104, USA. [10]These authors contributed equally: Lei Shen, Xiaokuang Ma, Yuanyuan Wang. ✉e-mail: sqiu@arizona.edu; xingyi@chop.edu; yyang@coh.org

diseases, including neurodevelopmental disorders and cancer[3–5]. Proteomic studies have identified thousands of arginine methylated protein substrates and the RNA-binding proteins (RBPs) constitute the largest group of these substrates[6–8]. Many of these RBPs are splicing factors and/or regulators of RNA splicing, thus, inhibiting arginine methylation using small molecule inhibitors or by the genetic knockout of PRMTs exerts profound impacts on RNA splicing[9–11]. However, PRMTs generally methylate a broad spectrum of protein substrates that are involved in various aspects of RNA metabolism, making it difficult to determine the specific function of arginine methylation in splicing regulation. Our previous work was the first to characterize the biochemical activity of PRMT9 and identified the splicing factor 3B subunit 2 (SF3B2) as its methylation substrate, linking the function of this newest member of the PRMT family to pre-mRNA splicing[12,13]. PRMT9 deposits SDMA marks at a single arginine site, R508. Our new data suggest that SF3B2 is the primary substrate of PRMT9 in human and mouse cells, and that R508 is highly methylated in vivo, indicating a conserved function of SF3B2 R508 methylation in regulating RNA splicing.

Alternative splicing (AS) is a fundamental regulatory mechanism of gene expression that allows the production of multiple mRNA isoforms from a single gene[14]. This process is tightly controlled by a large ribonucleoprotein complex known as the spliceosome[15–17]. The assembly of the spliceosome is guided by three main landmarks on the introns of pre-mRNA: the 5′ splice site (5SS), branch point sequence (BPS), and 3′ splice site (3′SS)[18–20]. These sequences function as core cis-elements that form multivalent RNA–protein and RNA–RNA interactions with the components of the spliceosome. In addition, other cis-regulatory elements located in introns and exons also act together with core splicing signals to ensure the fidelity of pre-mRNA splicing[14]. For example, the pre-mRNA anchoring site, which is a 20-nucleotide sequence located upstream of the BPS, can directly interact with several protein components of the U2 snRNP complex, including SF3B2, and facilitate spliceosome assembly[21,22]. AS is particularly prevalent in neuronal cells, where the splicing patterns are continuously changing to maintain cellular homeostasis and promote neurogenesis, migration, and synaptic function[23,24]. As such, genetic mutations that cause aberrations in neural AS patterns have been linked to many neurological diseases, including intellectual disability (ID), autism, and Alzheimer's disease[25]. Understanding the mechanisms of RNA splicing and the consequences of its disruption in neurological disorders is crucial for the development of potential therapies and treatments for these devastating conditions. Although recent advances in genome-wide technologies and the development of computational tools have facilitated the discovery of novel AS events at unprecedented levels[26–30], the molecular mechanisms underlying splicing regulation remain largely unknown.

In this study, we characterized a PRMT9 mutation, G189R, which is identified in autosomal recessive intellectual disability (ARID) in a large whole-genome sequencing study of 136 consanguineous families[31]. We found that this mutation completely abolishes PRMT9's catalytic activity on SF3B2 R508 methylation and leads to heavy PRMT9 ubiquitination. Based on the loss-of-function nature of this mutation, we established a *Prmt9* conditional knockout (cKO) mouse model and demonstrated that PRMT9 loss in excitatory neurons leads to aberrant synapse development and impaired learning and memory, thus revealing a causal relationship between PRMT9 loss-of-function and ARID. Importantly, using a complementary SF3B2 methylation-deficient knock-in (KI) mouse model, we showed that SF3B2 is the primary substrate of PRMT9 and that the neuronal defects observed in *Prmt9* cKO mice are likely due to SF3B2 methylation deficiency. Mechanistically, we found that R508 methylation modulates SF3B2–anchoring site interaction in a context-dependent manner, thereby influencing 3′ splice site selection.

## Results

### The ID patient-associated PRMT9 G189R mutation abolishes its arginine methyltransferase activity

PRMT9 harbors two essential AdoMet-binding domains with a clear duplication and partial conservation of most of the six signature PRMT motifs[13]. It also contains three tetratricopeptide repeats (TPRs) at its N terminus, potentially involved in protein–protein interactions (Fig. 1a). Amino-acid sequence analysis revealed that the G189R mutation identified in the ID patients is located within Motif I of the methyltransferase domain (Supplementary Fig. 1a). G189 is highly conserved among vertebrates (Fig. 1a) and among all human PRMTs (Supplementary Fig. 1a). Analysis of published human PRMT9 crystal structure (PDB: 6PDM) showed that G189 resides at the beginning of an α-helix located between the first and second β-strand of the characteristic seven-β-strand methyltransferase domain (Supplementary Fig. 1b). We conducted a series of biochemical and cellular assays to characterize this mutation. First, we compared the methyltransferase activity of G189R with wild type (WT) and a previously reported catalytic-inactive mutant PRMT9 (VLDI mutated to AAAA; "4A")[13] using an in vitro methylation assay. Recombinant G189R failed to methylate SF3B2, the only reported PRMT9 substrate (Fig. 1b). Second, we compared the interaction of WT and mutant PRMT9 with SF3B2 by performing a co-immunoprecipitation (co-IP) assay. Like the catalytic-inactive 4A mutant, G189R failed to interact with SF3B2 in cells (Fig. 1c). Third, using the SF3B2 site-specific arginine methylation antibody (αSF3B2 R508me2s) that we developed in our previous study, we found that only the rescue expression of WT PRMT9, but not the G189R or 4A mutant, restored SF3B2 R508me2s level in PRMT9 KO cells (Fig. 1d). Lastly, immunofluorescence (IF) staining assay using the SF3B2 R508me2s antibody showed that the G189R and 4A mutations did not affect PRMT9 cytoplasmic localization, but neither of them could restore SF3B2 R508me2s in PRMT9 KO cells (Fig. 1e). Altogether, these results demonstrate that the ID patient-associated G189R mutation abolishes the arginine methyltransferase activity of PRMT9 in vitro and in cells.

### G189R-mutant PRMT9 protein is unstable and subject to ubiquitination by UBE3C

The G189R mutant is not only catalytically inactive but also very unstable. When we transfected cells with equal amounts of WT and G189R-mutant PRMT9 plasmids, which resulted in similar levels of mRNA expression, the protein level of the G189R mutant was dramatically lower than that of WT PRMT9 (Supplementary Fig. 2a), suggesting that the G189R mutation may affect PRMT9 protein stability. We tested this possibility by comparing the half-lives of WT and G189R-mutant PRMT9. Although WT PRMT9 did not exhibit obvious degradation within the treatment period (8 h), the G189R-mutant protein was quickly degraded, with a half-life of 1–2 h, similar to that of the well-known short half-life protein myeloid cell leukemia-1 (MCL-1). Importantly, treating cells with the proteasome inhibitor MG132 significantly stabilizes the G189R-mutant protein (Supplementary Fig. 2b), suggesting that it is likely subject to the ubiquitin-proteasome pathway for degradation. To further test this, we checked for the ubiquitin modification of both WT and G189R-mutant PRMT9 in an in vitro ubiquitination assay. Compared to WT PRMT9, the G189R-mutant protein was heavily ubiquitinated, as evidenced by a strong smear signal of poly-ubiquitinated mutant protein (Fig. 2a). This effect was further exaggerated when HEK293T cells were transfected with HA-ubiquitin and treated with the proteasome inhibitor MG132. The heavily ubiquitinated G189R-mutant protein could even be detected in the input samples, which is evidenced by a strong non-resolvable smear of poly-ubiquitinated G189R-mutant protein retained in the stacking gel (Fig. 2b). Altogether, these results demonstrate that G189R-mutant PRMT9 is unstable and is heavily ubiquitinated in cells.

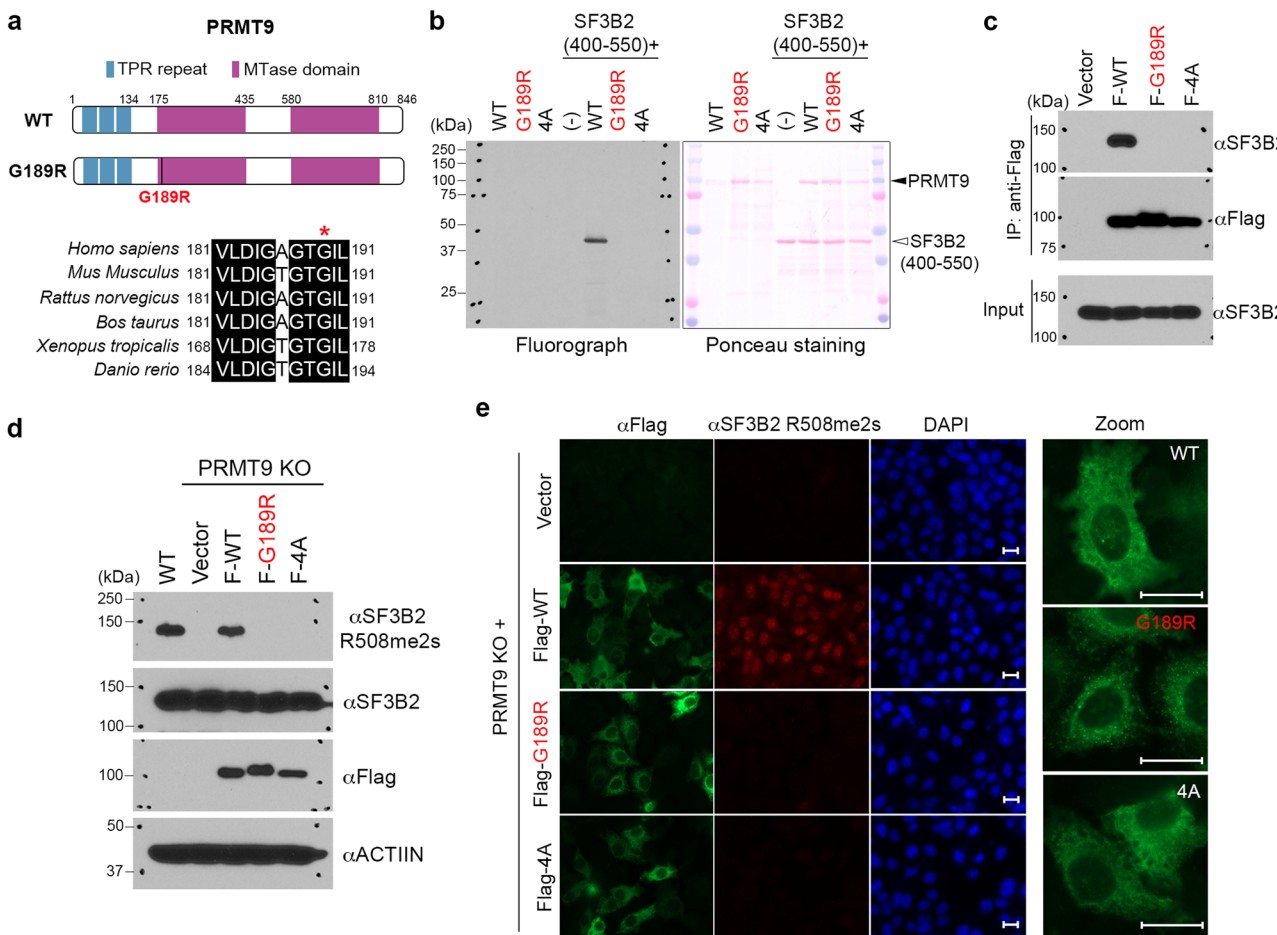

**Fig. 1 | The ID patient-associated PRMT9 G189R mutation abolishes its arginine methyltransferase activity. a** Domain structure of human PRMT9. The N-terminal TPR repeats are in blue, and the two tandem methyltransferase (MTase) domains are in purple. The ID-associated G189R mutation, which is located at the N-terminus of the first methyltransferase domain, is highlighted in red. G189 (red asterisk) is conserved in PRMT9 genes ranging from zebrafish to human. **b** G189R-mutant PRMT9 is catalytically inactive. In vitro methylation assay was performed by incubating WT, G189R-mutant, and catalytic-inactive (4A) recombinant PRMT9 enzymes with a fragment of SF3B2 (a.a. 400–550). 4A, catalytic-inactive PRMT9 mutant (VLDI to AAAA). n = 3; n is number of independent experiments unless otherwise stated. **c** G189R-mutant PRMT9 fails to interact with its methylation substrate SF3B2. A co-IP assay was performed to detect the interactions of Flag-tagged WT, G189R-mutant, and catalytic-inactive (4A) PRMT9 with the endogenous SF3B2 (n = 3). **d** G189R-mutant PRMT9 fails to restore SF3B2 arginine methylation (SF3B2 R508me2s) in PRMT9 KO cells. Western blot analysis was performed to detect SF3B2 R508 methylation level in WT and PRMT9 KO HeLa cells with rescue expression of Flag-tagged WT, G189R-mutant, and 4A-mutant PRMT9 (n = 3). **e** The G189R mutation does not affect the overall cytoplasmic localization of PRMT9, but G189R-mutant PRMT9 fails to catalyze SF3B2 arginine methylation, consistent with the results in (**d**). Immunofluorescence assay was performed to detect SF3B2 R508 methylation level (red) in WT and PRMT9 KO HeLa cells with rescue expression of Flag-tagged WT, G189R-mutant, and 4A-mutant PRMT9 (green). α-Flag demonstrates the subcellular localization of the Flag constructs (n = 3). Scale bar: 20 μm. Source data are provided as a Source Data file.

To identify the E3 ubiquitin ligase that catalyzes PRMT9 ubiquitination, we performed liquid chromatography–mass spectrometry (LC/MS) analysis on protein complexes co-purified with Flag-tagged WT and G189R-mutant PRMT9 (Fig. 2c). Consistent with our previous report and the results shown in Fig. 1c, we found that SF3B2 and SF3B4 co-purified with WT but not G189R-mutant PRMT9 (Fig. 2d, black font). Importantly, we identified the ubiquitin-protein ligase E3C (UBE3C) and valosin-containing protein (VCP/p97) as two novel PRMT9-interacting proteins (Fig. 2d, blue font). Surprisingly, higher numbers of unique peptides from UBE3C and VCP/p97 were identified in the G189R-mutant protein complex (Fig. 2d), suggesting that UBE3C and VCP/p97 prefer to interact with G189R-mutant PRMT9. Indeed, in a co-IP validation experiment, we found that the interactions between UBE3C and VCP/p97 with G189R-mutant PRMT9 were much stronger than that of WT PRMT9 (Fig. 2e). UBE3C is a cytosolic HECT (homologous to the E6-AP carboxyl terminus) domain-containing E3 ubiquitin ligase[32], and VCP/p97 functions as a molecular chaperone by guiding ubiquitinated protein substrates to the 26S proteasome for degradation[33]. Thus, UBE3C and VCP/p97 could act together to mediate PRMT9 degradation. We focused on UBE3C and tested the hypothesis that UBE3C catalyzes PRMT9 ubiquitination. Supporting this hypothesis, knockdown of UBE3C using siRNA increased PRMT9 expression, whereas overexpression of Flag-tagged UBE3C reduced PRMT9 expression (Fig. 2f). The RNA level of PRMT9 was not affected by overexpression or knockdown of UBE3C (Supplementary Fig. 2c, d). Furthermore, UBE3C knockdown significantly increased PRMT9 protein stability (Fig. 2g). Particularly, the half-life of G189R-mutant PRMT9 was extended from 1–2 h to more than 8 h (the last time point in this experiment). As a control, the stability of MCL-1 was not affected by UBE3C knockdown (Fig. 2g). To define UBE3C as the E3 ubiquitin ligase for PRMT9 ubiquitination, we performed in vivo and in vitro ubiquitination assays using WT and E3 ligase-deficient mutant UBE3C (C1051S)[32]. Overexpression of WT UBE3C, but not the C1051S mutant, dramatically enhanced PRMT9 ubiquitination (Fig. 2h). As shown in the in vitro ubiquitination assay, only WT recombinant UBE3C, but

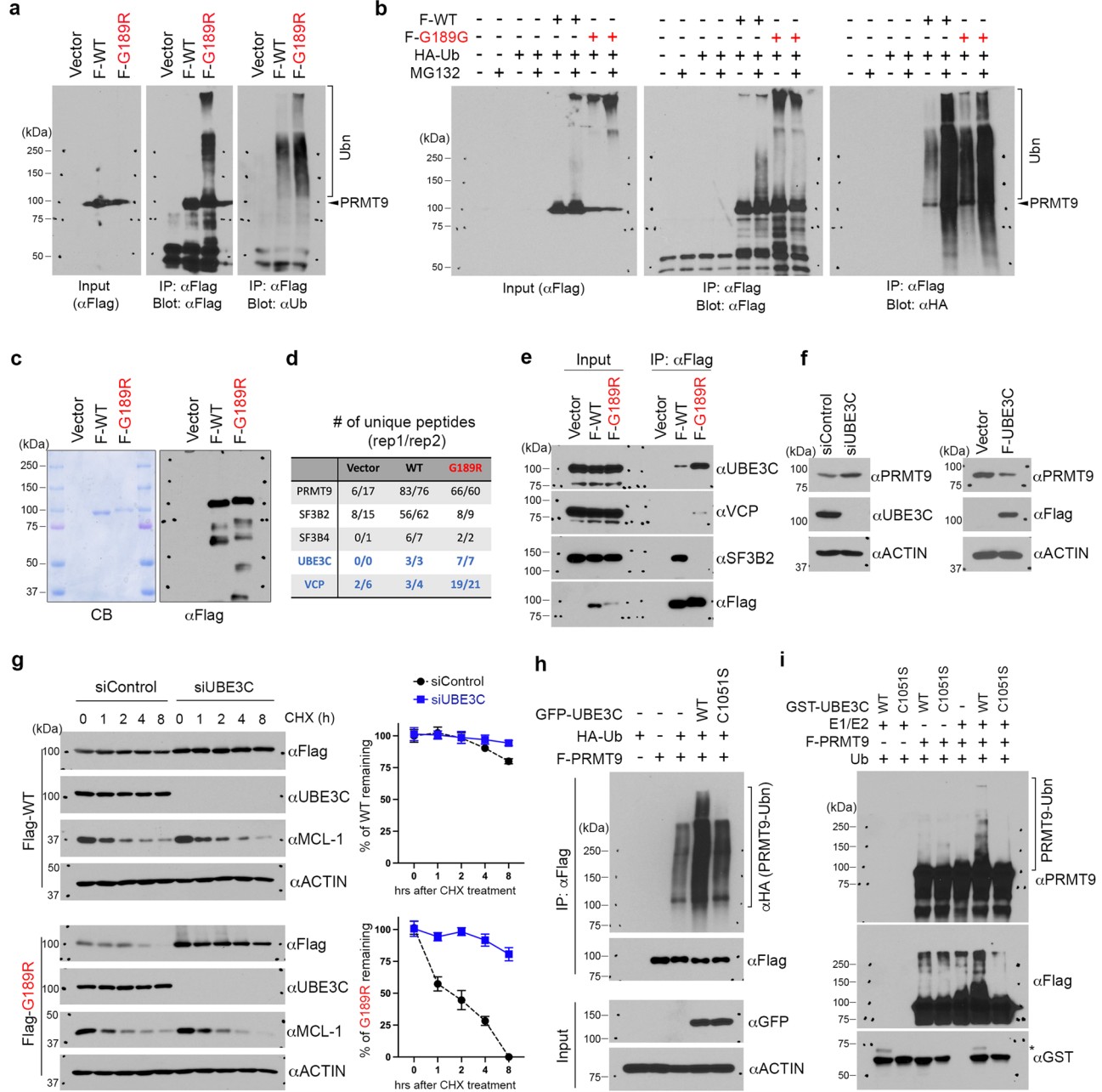

**Fig. 2 | G189R-mutant PRMT9 is unstable and subject to ubiquitination by UBE3C. a** G189R-mutant PRMT9 protein is heavily ubiquitinated in cells. Flag-tagged WT or G189R-mutant PRMT9 were immunoprecipitated. Samples were detected by western blot using αFlag and αUb antibodies (n = 3). **b** Detection of PRMT9 ubiquitination using in vivo ubiquitination assay. HEK293T cells were transfected with Flag-tagged WT or G189R-mutant PRMT9 together with HA-tagged Ubiquitin and treated with the proteasome inhibitor MG132 before harvest (n = 3). **c** Purification of WT and G189R-mutant PRMT9 protein complex from HEK293 cells. The purified protein complex was visualized by Coomassie blue staining (left panel) and an αFlag western blot (right panel). n = 2. **d** Identification of PRMT9 protein complex by liquid chromatography–mass spectrometry (LC/MS) with two biological repeats (rep1 and rep 2). The number of unique peptides from individual proteins is listed. Known PRMT9 interaction proteins are labeled in black and novel interaction proteins are labeled in blue. **e** Validation of PRMT9 interaction proteins by co-immunoprecipitation. Flag-tagged WT or G189R-mutant PRMT9 were immunoprecipitated with an αFlag M2 magnetic beads, and the protein complexes were detected by western blot (n = 3). **f** UBE3C negatively regulates PRMT9 protein level. HeLa cells were transfected with UBE3C siRNA or Flag-tagged UBE3C to either knockdown or overexpress UBE3C (n = 3). **g** Knockdown UBE3C stabilizes PRMT9 protein. Protein stability of Flag-tagged WT or G189R-mutant PRMT9 was determined in control and UBE3C knockdown HeLa cells (left panel). PRMT9 western blot signal was quantified using ImageJ software (right panel). Data are presented as mean values ± SD. Error bars represent standard deviation calculated from three independent western blots. **h** UBE3C catalyzes PRMT9 ubiquitination in cells. In vivo ubiquitination assay was performed with GFP-tagged wild type (WT) and catalytic-deficient (C1051S) UBE3C (n = 3). **i** UBE3C catalyzes PRMT9 ubiquitination in vitro. In vitro ubiquitination assay was performed using indicated recombinant proteins (n = 3). * UBE3C auto-ubiquitination. Source data are provided as a Source Data file.

not the C1051S mutant, could catalyze PRMT9 ubiquitination in the presence of E1/E2 and ubiquitin (Fig. 2i). Altogether, these results demonstrate that UBE3C is an E3 ubiquitin ligase that catalyzes PRMT9 ubiquitination.

## *Prmt9* whole-body knockout affects postnatal mouse development

Using biochemical and cellular assays, we showed that the G189R mutation completely abolishes PRMT9 arginine methyltransferase

activity (Fig. 1), and dramatically shortens the half-life of PRMT9 protein due to heavy ubiquitination (Fig. 2), strongly suggesting that this mutation is a loss-of-function mutation. On this basis, we reasoned that a *Prmt9* conditional knockout (cKO) mouse model would provide valuable information about the molecular basis underlying the pathogenesis of ID caused by the G189R mutation. Thus, we established a *Prmt9* conditional allele (*Prmt9*flox) in mouse, in which *Prmt9* exon 5 is flanked by *loxP* sites (Supplementary Fig. 3a). Cre-mediated exon 5 deletion would create a frameshift and premature stop codon, truncating a large portion of the methyltransferase domain (Supplementary Fig. 3b, c).

Because the mouse monoclonal PRMT9 antibody we developed in the previous study[13] does not recognize mouse PRMT9 protein, to facilitate the detection of mouse PRMT9 protein, we generated two rabbit-polyclonal PRMT9 antibodies (namely #97 and #98). The specificity of these antibodies was validated by western blot using total cell lysates extracted from *Prmt9* KO mESCs and HeLa cells (Supplementary Fig. 3d). In both cell lines, loss of PRMT9 completely abolished SF3B2 R508me2s. Notably, although both #97 and #98 recognize mouse PRMT9 protein in the WT, but not in the KO, lysates, neither cross-reacts with human PRMT9 protein (Supplementary Fig. 3d). To investigate the impact of *Prmt9* loss on mouse development, we deleted *Prmt9* in the germline using a *CMV*Cre transgenic mouse line[34]. Using the polyclonal PRMT9 antibody, we confirmed the complete loss of PRMT9 protein and SF3B2 R508me2s in all tissues of *Prmt9* KO mice (Supplementary Fig. 3e). Although we did not observe gross abnormalities during embryonic development, PRMT9 loss did cause partial postnatal lethality (Supplementary Fig. 3f), and the KO mice were smaller (Supplementary Fig. 3g) and weighed less (Supplementary Fig. 3h), with no obvious lean/fat mass differences (Supplementary Fig. 3i, j). These results indicate that PRMT9 is involved in postnatal development.

## *Prmt9* knockout in excitatory neurons impairs mouse learning and memory

To understand how PRMT9 loss-of-function is related to ID, we decided to investigate its function in the mouse brain. In situ hybridization data from the Allen Brain Atlas shows that *Prmt9* mRNA is widely expressed across many telencephalon structures, including the prefrontal cortex (PFC) and hippocampus (Supplementary Fig. 4a). We detected PRMT9 protein expression in micro-dissected hippocampus CA1 regions of postnatal day 0 (P0) to P90 mice and found that PRMT9 expression peaks around P7-P14 and declines after P21 (Supplementary Fig. 4b), indicating that PRMT9 is tightly regulated and plays a role in early postnatal brain development. To further dissect the cell-type specific role of PRMT9 and rule out potential confounding effects of whole-body knockout, we evaluated behaviors related to learning and memory using a *Prmt9*fx/fx:*emx1*cre cKO mouse model, in which *Prmt9* is deleted in cortical excitatory neurons[35,36]. Using the Morris water maze (MWM) test, a standard assay to assess hippocampus-dependent spatial memory (Fig. 3a), we found significantly slower spatial memory acquisition in the *Prmt9* cKO mice (10-14 weeks age) during the MWM training phase (Fig. 3b–d). During the probe trial, the *Prmt9* cKO mice also spent less time in the target quadrant (Fig. 3e). In the reverse learning test, during which the platform was moved to a new location, cKO mice also exhibited slower learning of the new platform location, indicating impaired cognitive flexibility (Fig. 3f, g). We next conducted a Pavlovian fear conditioning test to further assess associative memory, which is broadly dependent on cortical functions (Fig. 3h). The *Prmt9* cKO mice were slower in associating an auditory tone (conditioned stimulus) with a foot shock (unconditioned stimulus) (Fig. 3i). In addition, these mice showed less conditioned contextual freezing during the test phase (Fig. 3j). Because *Prmt9* is broadly expressed in the developing mouse cortex, including the motor cortex (Supplementary Fig. 4a), we assessed the locomotor activity and anxiety-related behaviors in *Prmt9* cKO and control littermate mice. Using an open field (OF) activity chamber and an elevated plus maze (EPM) (Supplementary Fig. 4c, f), we found no significant differences between the two groups in terms of the total distance traveled in OF (Supplementary Fig. 4d), the percent of time spent in center of the arena in OF (Supplementary Fig. 4e), the time spent in the open-arm of the EPM (Supplementary Fig. 4g), or in the total number of open arm entries (Supplementary Fig. 4h). These data indicate normal basal locomotor activity, and similar levels of anxiety or reaction to novelty between *Prmt9* cKO and control mice. Taken together, these results demonstrate that PRMT9 loss-of-function in developing cortical circuits impairs learning, memory, and cognitive flexibility.

## *Prmt9* cKO hippocampus neurons show impaired excitatory synapse development

Our behavioral analysis revealed that *Prmt9* cKO mice exhibited impaired learning and memory (Fig. 3). To determine the cellular basis underlying these impairments, we assessed the morphology of hippocampus CA1 neurons in *Prmt9* cKO mice by using biocytin filling after patch clamp recordings[37,38]. While we did not observe significant effects of *Prmt9* deficiency on dendritic arborization (Fig. 4a), *Prmt9* cKO neurons showed a significant reduction in spine density and head volume (Fig. 4b–d), indicating defective synapse development. Next, we investigated how *Prmt9* deficiency affects excitatory synapse maturation by culturing primary embryonic hippocampus neurons from WT and *Prmt9* cKO mice. We quantified functional synapses by co-staining the AMPA/NMDA glutamate receptor subunits (GluA1/GluN1) and pre-/post-synaptic markers (PSD95/Synapsin I) (Fig. 4e). *Prmt9* cKO neurons showed a reduced density of PSD95+, GluN1+, and GluA1+ puncta and decreased colocalization of pre-/post-synaptic and AMPA/NMDA markers (Fig. 4f). Note that these results were obtained from in vitro low density cultured hippocampal neurons grown in conditions independent of extensive in vivo network activities and neuron–glia interactions. Thus, the results indicate an important cell autonomous effect of *Prmt9* loss-of-function in reducing the number of functional excitatory synapses.

We next tested the effects of *Prmt9* loss in vivo by measuring spontaneous miniature excitatory postsynaptic currents (mEPSCs) in CA1 neurons from young adult (12–14 weeks old) *Prmt9* cKO mice and their littermate controls (Fig. 4g). Whole-cell patch clamp recording revealed that cKO CA1 neurons showed reduced mEPSC amplitude, with more mEPSCs amplitudes clustered to the lower amplitude bins (Fig. 4h, $p = 0.015$, Kolmogorov-Smirnov test). mEPSC frequency was also decreased in cKO CA1 neurons compared to WT neurons (Fig. 4i, $t29 = 3.11$, $p = 0.004$). In addition, a decreased AMPA/NMDA receptor current ratio was observed in cKO CA1 neurons (Fig. 4j, $t19 = 2.96$, $p = 0.008$). These functional measures are consistent with the observed decreased CA1 neuron dendritic spine density and head sizes (Fig. 4b–f) and suggest impaired excitatory synapse maturation as a result of *Prmt9* loss. Based on the observed learning and memory deficits (Fig. 3), we next conducted field potential recording and probed synaptic transmission and plasticity (long-term potentiation/depression, LTP/LTD), which are putative synaptic underpinnings for learning and memory[39,40]. We found an overall reduction in postsynaptic fEPSP response to presynaptic inputs in cKO CA1 responses (Fig. 4k. Test on slope difference, $F = 9.4$, $p = 0.003$), while paired pulse ratio responses across various inter-stimulus intervals was not altered (Fig. 4l, $F(1,18) = 0.57$, $p = 0.46$, for group effects). Lastly, quantification of the averaged last 10-min LTP amplitude revealed a significantly reduction of LTP from cKO CA1 region (Fig. 4m, $t18 = 10.4$, $p < 0.0001$). Similarly, a reduced level of LTD was also observed (Supplementary Fig. 4i, j). These data strongly suggest a critical role of PRMT9 in synapse development and postnatal synaptic plasticity and underscore the neurobiological basis for the

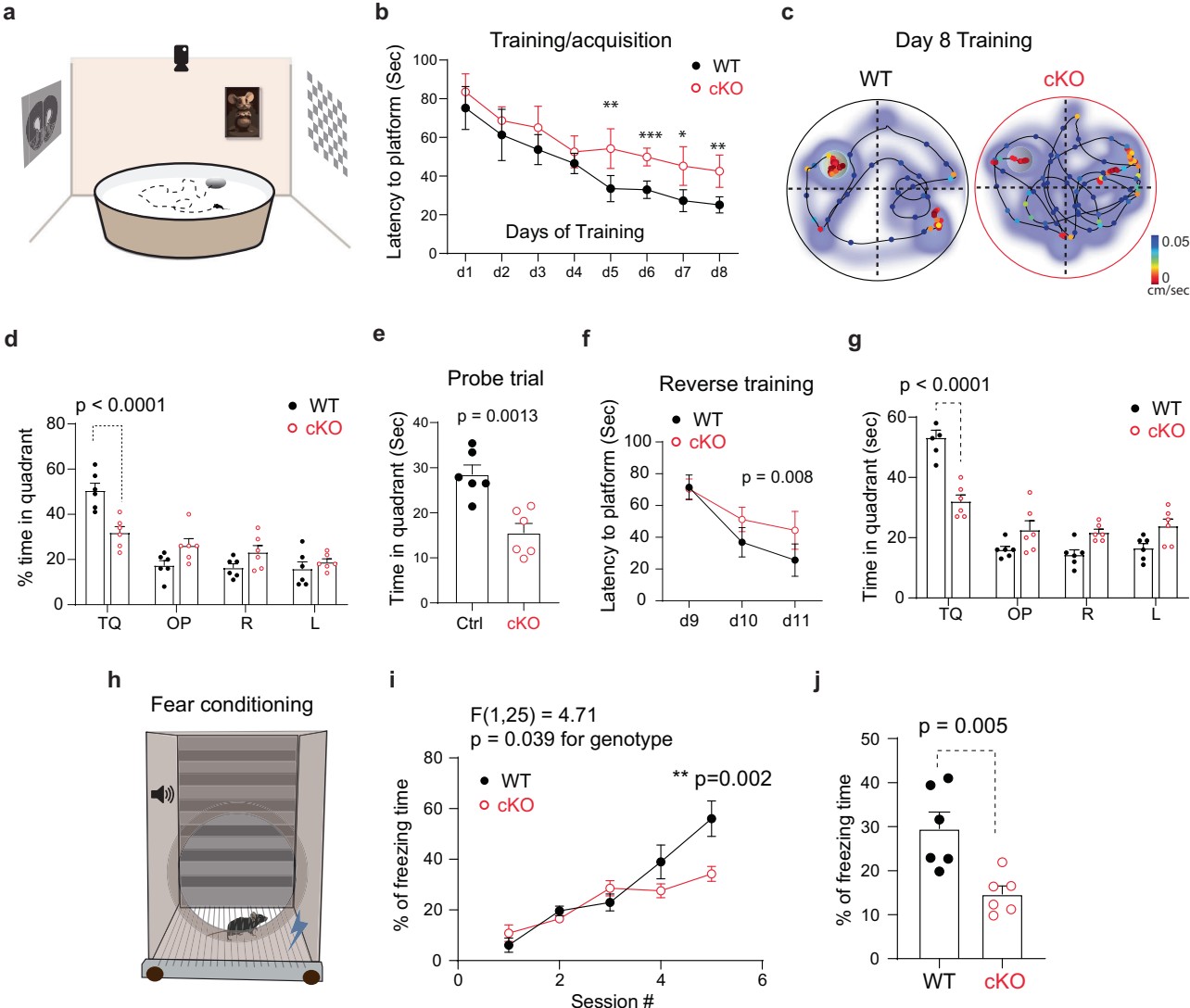

**Fig. 3 | *Prmt9* cKO mice show impaired learning and memory. a** Graphic illustration of Morris Water Maze test, with prominent ambient visual cues surrounding the test arena to facilitate associative learning. **b** *Prmt9* cKO mice showed slower learning during the acquisition period. The y-axis shows the time taken to find the hidden platform during the eight training days. Data are presented as mean ± SEM. Main effect on groups, $F (1,12) = 19.4$, $p = 0.0009$. *$p = 0.019$, **$p = 0.009$, ***$p < 0.001$. Repeated measures two-way-ANOVA with Sidak's multiple comparison test. **c** Representative mouse moving tracks and speed on Day 8 of training. Heatmap, time spent at location of arena; line graph, moving trajectory of mice. Marker color denotes swimming speed at the corresponding location. **d** Quantification of percent time spent in each quadrant during Day 8 of training. *Prmt9* cKO mice spent significantly less time ($p < 0.0001$, unpaired $t$ test) in the target quadrant (TQ). Data are presented as mean ± SEM. **e** *Prmt9* cKO mice showed impaired memory. *Prmt9* cKO mice spent significantly less time in the target quadrant during the probe trial ($p = 0.0013$, unpaired $t$ test). Data are presented as mean ± SEM. **f** *Prmt9* cKO mice showed slower learning of the new platform location during Day 10–12 reverse

learning period. Data are presented as mean ± SEM. $p = 0.008$ for the effect of genotype, repeated measures two-way rmANOVA. **g** *Prmt9* cKO mice spent less time in the new target quadrant during the Day 11 of the probe trial ($p < 0.0001$, unpaired $t$ test). Data are presented as mean ± SEM. Two-way rmANOVA with Sidak's MCT. For (**a**–**g**), WT, n = 6 (4M2F); cKO, n = 6 (3M3F). **h** Schematic diagram of a Pavlovian fear conditioning paradigm. A different cohort of mice was used. WT, n = 6 (3M3F); cKO, n = 6 (2M4F). Day 1 had five training sessions in which a mild foot shock (US) was paired with an auditory cue (CS). On day 2, mice were placed in the same context with no US or CS stimulus presented. Freezing time was measured. **i** *Prmt9* cKO mice showed decreased fear conditioning learning. *Prmt9* cKO mice exhibited a significantly decreased freezing time and different time course in associating the unconditional (audio cue) stimulus with foot shock and freezing behavior compared with controls, $p = 0.04$, for the effects of genotype, repeated measures two-way ANOVA. Data are presented as mean ± SEM. **j** *Prmt9* cKO mice show reduced contextual-induced freezing behavior ($p = 0.005$, unpaired $t$ test). Data are presented as mean ± SEM. For (**h**–**j**), WT, n = 6 (3M3F); cKO, n = 6 (2M4F).

---

impaired learning/memory and cognition seen with the ID-associated G189R mutation.

## SF3B2 is the primary substrate of PRMT9 in human cell lines and mouse tissues

We previously identified SF3B2 as an interaction partner and methylation substrate of PRMT9[12,13]. Using native total lysate from HeLa cells as methylation substrates, we found that recombinant PRMT9 only methylates one predominant band that corresponds to the SF3B2

protein in *Prmt9* KO, but not WT lysates, whereas PRMT1 methylates a multitude of cellular proteins in both control and *Prmt9* KO cells (Supplementary Fig. 5a). The $^3$H-labeled methylation signal was indeed from SF3B2, because the methylation signal was drastically diminished upon SF3B2 knockdown by siRNA (Supplementary Fig. 5b). We further confirmed that recombinant PRMT9 can methylate SF3B2 only in lysates from *Prmt9* KO mESCs (Supplementary Fig. 5c) and various tissues of *Prmt9* KO mice (Supplementary Fig. 5d), including hippocampus (Fig. 5a), strongly suggesting that SF3B2 is the primary

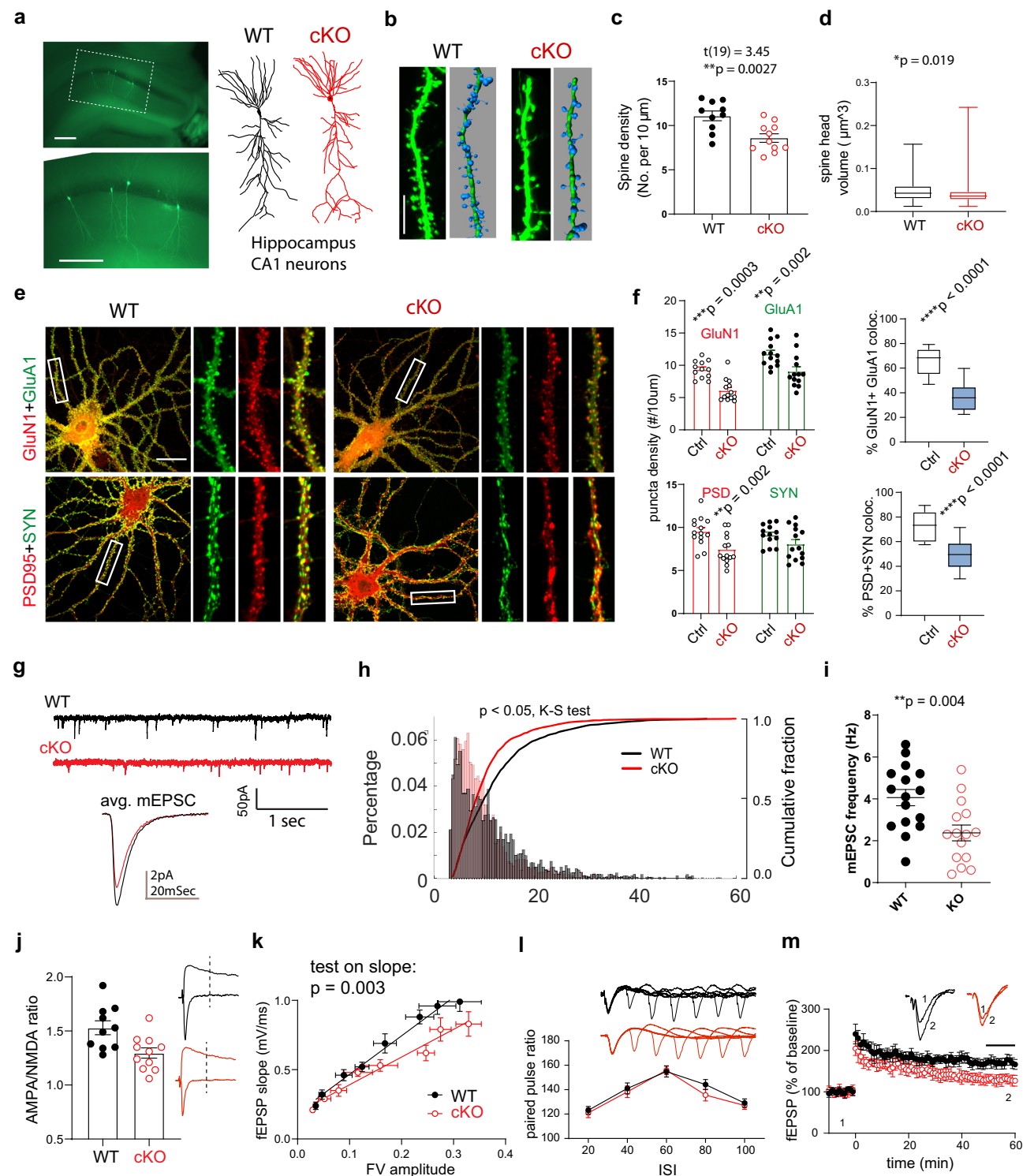

substrate of PRMT9. Note that recombinant PRMT9 fails to catalyze additional SF3B2 methylation in WT cells, suggesting that R508 is methylated at a high level in vivo. A recent study reported that the mitochondrial antiviral-signaling protein (MAVS) can be methylated by PRMT9[41]; however, we were unable to detect MAVS methylation under the same assay conditions that detect strong SF3B2 methylation (Supplementary Fig. 5e). All these results strongly support that SF3B2 R508 is the major, if not only, PRMT9 substrate. Thus, we hypothesize that the impaired synaptic development and function in *Prmt9* cKO mice result from dysregulated pre-mRNA splicing due to the lack of SF3B2 methylation at R508.

## PRMT9 regulates pre-mRNA splicing of genes involved in synaptic functions

To define the role of PRMT9-regulated RNA splicing in synapse development and function, we performed RNA-seq analyses on RNA samples extracted from hippocampus tissues of two-week-old *Prmt9* KO female mice and their same-sex littermate controls (three biological replicates per genotype), particularly focusing on changes in alternatively spliced events. For RNA-seq, we performed 100 bp paired-end sequencing and achieved more than 200 million unique mapped reads for each sample (Supplementary Fig. 6a). Reads from the floxed exon 5 were completely abolished, as expected, and the

**Fig. 4 | *Prmt9* cKO hippocampus neurons show impaired excitatory synapse development. a** Photomicrograph of CA1 hippocampus neurons and their representative dendritic arborization. Scale bar, 200 μm. **b** Representative apical dendritic segments with reconstructed dendritic spines. Scale bar, 10 μm. **c** *Prmt9* cKO CA1 neurons (n = 11) showed reduced dendritic spine density compared to that from WT (n = 10) neurons ($p = 0.0027$, unpaired *t* test). Data are presented as mean ± SEM. **d** *Prmt9* cKO mice showed reduced dendritic spine head volume/size in CA1 neurons (cKO, n = 141 spines/7 neurons; WT, n = 133 spines/8 neurons. $p = 0.019$, Kolmogorov-Smirnov test). Box plot indicates min, lower quartile Q1, median, upper quartile Q3 and max. **e** Representative co-labeling of synapse markers, including the glutamate receptor subunits (GluA1/GluN1) and the pre- and post-synaptic proteins (Synapsin I/PSD95) in WT control and *Prmt9* cKO mice (n = 3). Scale bar, 20 μm. **f** Quantification of GluA1/GluN1 and Synapsin I/PSD95 puncta density and double labeling. Cultured *Prmt9* cKO hippocampal neurons show reduced puncta density for GluN1 ($p = 0.0003$) and GluA1 ($p = 0.002$) compared to WT/control neurons (two-way ANOVA with Sidak's MCT). In addition, cKO hippocampal neurons show reduced density for PSD95 puncta ($p = 0.002$). Reduced proportion of functional excitatory synapses, defined by the proportion of colocalized GluA1/GluN1 ($p < 0.0001$, n = 13 neurons for control and cKO) and Synapsin I/PSD95 ($p < 0.0001$, n = 14 neurons for control and cKO) was observed in cKO neurons as well. Box plot indicates min, lower quartile Q1, median, upper quartile Q3 and max. **g** Exemplary whole cell mEPSC traces from WT and *Prmt9* cKO CA1 neurons. **h** Cumulative and percentage distribution of mEPSC amplitudes from WT control and *Prmt9* cKO CA1 neurons. Prmt9 cKO neurons showed decreased mEPSC amplitude ($p = 0.015$, Kolmogorov–Smirnov test). **i** *Prmt9* cKO CA1 neurons show reduced mEPSC frequency (WT, n = 16 neurons; cKO, 15 neurons. $p = 0.004$). **j** *Prmt9* cKO CA1 neurons exhibit significantly smaller AMPA/NMDA current ratio (WT, n = 10 neurons; cKO, n = 11 neurons. $p = 0.02$). Data are presented as mean ± SEM. **k** Input–output responses in CA1 field potential recordings, measured by fEPSP slope as a function of fiber volley amplitude (WT, n = 9 slices; cKO, n = 8 slices. $p = 0.003$). **l** Paired pulse responses across 20–100 ms ISI as a measure of presynaptic function (WT, n = 10 slices; cKO, n = 10 slices. $p = 0.46$ for the effects of genotype). **m** *Prmt9* cKO hippocampus CA1 show a significantly lowered LTP magnitude, measured as the last 10-min responses (WT, n = 10 slices; cKO, n = 10 slices. $p < 0.0001$ for the effects of genotype).

reads from other exons were also dramatically reduced, likely because of the nonsense-mediated mRNA decay (Supplementary Fig. 6b). In total, 3722 alternative splicing events from 1953 genes were significantly altered upon *Prmt9* deletion, including exon skipping (SE), alternative 5′ or 3′ splice sites (A5SS or A3SS), mutual exclusive exons (MXE), and intron retention (RI) (Fig. 5b). In particular, A3SS were over-represented among the overall alternatively spliced events (Supplementary Fig. 6c), indicating a role of PRMT9-catalyzed SF3B2 methylation in 3′ splice site (3′SS) selection. To our surprise, no differentially expressed genes (except for *Prmt9* itself) were identified (fold change > 1.5 and FDR < 0.01) between control and *Prmt9* KO hippocampus (Fig. 5c), suggesting that PRMT9 functions in regulating isoform-specific gene expression. A similar observation has been reported when the neuronal splicing factor Nova2 was knocked out in neocortex[42].

To determine the extent to which the dysregulation of RNA splicing might contribute to impaired synaptic function in *Prmt9* KO mice, we performed Gene Ontology (GO) and pathway analysis on *Prmt9*-regulated splicing targets, particularly SE and A3SS. These analyses revealed a strong enrichment of alternatively spliced transcripts in synapse-related pathways, including signal transmission and the N-methyl-D-aspartate (NMDA) receptor activation (Fig. 5d and Supplementary Fig. 6d). Importantly, many of the genes critical for synaptic functions are targets of *Prmt9*-regulated splicing, such as *Stxbp5l* (Syntaxin-binding protein 5-like), *Grin1* (Glutamate Ionotropic Receptor NMDA Type Subunit 1), and Rap1gap (Rap1 GTPase-activating protein 1) (Fig. 5e and Supplementary Fig. 6e). Surprisingly, among the 73 causative variants of ARID identified by Najmabadi et al[31]., 10 are direct targets of PRMT9-regulated splicing, and additional 28 have paralogs that are regulated by PRMT9 (Supplementary Fig. 6f). Although ARID is highly heterogeneous, we found that 27 of the 38 protein components of the Ras/Rho/PSD95 network[31,43] are splicing targets of PRMT9 (Supplementary Fig. 6g) and that PRMT9-regulated alternatively spliced genes are more enriched in postsynapse components (Supplementary Fig. 6h), suggesting that PRMT9 might regulate synaptic functions through Ras and Rho GTPase signaling pathway, which includes critical factors controlling dendritic spine morphogenesis and neural circuit connectivity[44–47].

**Loss of SF3B2 methylation resembles *Prmt9* KO in causing defective synaptic function and hippocampus RNA splicing**
We showed that SF3B2 is the primary methylation substrate of PRMT9 (Supplementary Fig. 5). To further assess the extent to which the effects of *Prmt9* KO on mouse behavior and RNA splicing are caused by the loss of SF3B2 methylation, we generated an Sf3b2 arginine methylation-deficient mutant mouse model by substituting the methylated arginine

residue to lysine (K), namely Sf3b2$^{R491K}$ (corresponding to human R508) (Supplementary Fig. 7a). The homozygous R491K knock-in (KI) mice exhibited a complete loss of PRMT9-mediated arginine methylation without affecting the expression of PRMT9 (Supplementary Fig. 7b, c). Similar to *Prmt9* KO, R491K mice showed partial postnatal lethality (Supplementary Fig. 3f), and most importantly, they also exhibited impaired learning and memory when tested using the Morris water maze at ~2 months of age (Supplementary Fig. 7d–g). Consistently, field potential recording revealed an impaired LTP at the CA3>CA1 synapse in these mice (Supplementary Fig. 7h, i). Furthermore, we performed RT-PCR and analyzed RNA splicing of PRMT9 targets in hippocampus from WT and R491K mice. All 9 differentially spliced genes identified in *Prmt9* KO showed similar splicing changes in R491K hippocampus (Supplementary Fig. 7j). Together, these results strongly suggest that the biological function of PRMT9 in neurodevelopment and RNA splicing is mediated by its activity in methylating SF3B2.

**PRMT9-regulated alternative splicing exhibits weaker 3′ splice site features, including nonconsensus anchoring site sequences**
The cis-regulatory elements that are essential for pre-mRNA splicing consist of the 5′ (GU) and 3′ (AG) splice site, BPS, and polypyrimidine track located upstream of the 3′ splice site[48]. Additionally, the sequence located 6–25 nt upstream of the branch point, called anchoring site[21,49], is also involved in the sequence-dependent binding of the SF3B complex in spliceosome assembly (Fig. 6a). To determine the sequence features of alternative splicing (AS) events regulated by PRMT9, we compared the intron and exon sequence features near the differentially spliced cassette exons (SE) in *Prmt9* KO hippocampus to the transcriptome-wide background cassette exons. First, we examined the splice site strength of the two 5′ splice sites and the two 3′ splice sites involved in defining SE events. Although no significant differences were observed when comparing the 5′ splice sites located at either upstream or downstream intron (Fig. 6b, left panel), we found that cassette exons that were more included or excluded upon *Prmt9* KO were associated with weaker 3′ splice sites in upstream and downstream introns, respectively (Fig. 6b, right panel). Next, we extracted sequences of introns upstream and downstream of the cassette exons to predict BPS using the Branch Point Prediction software[50]. The branch point predicted in introns upstream of more included cassette exons are more distal to their corresponding 3′ splice sites (Fig. 6c) and have lower predicted branch point sequence scores (Fig. 6d). Importantly, as shown in the nucleotide frequency logos near the predicted branch point adenosine (A), more nonconsensus sequences were observed within the anchoring site in upstream or downstream introns for cassette exons that are more included and excluded upon *Prmt9* KO,

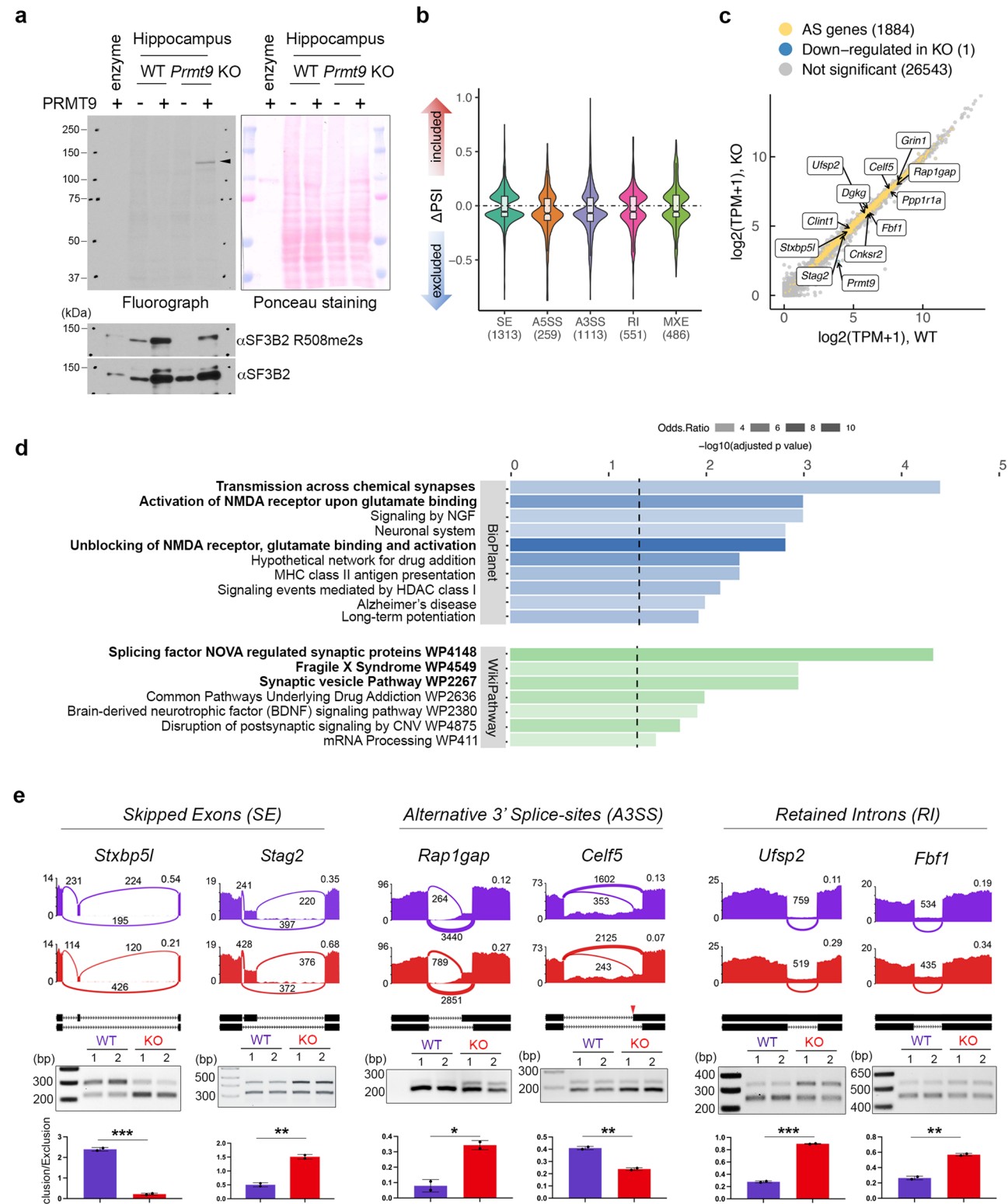

respectively (Fig. 6e), suggesting that the splicing of exons with nonconsensus anchoring site sequences is more likely to be regulated by PRMT9.

## PRMT9 regulates RNA splicing through fine-tuning the interaction of SF3B2 with pre-mRNA anchoring site

We previously showed that R508 methylation promotes the interaction of SF3B2 with the survival of motor neuron (SMN) and potentially

U2 snRNP maturation[13]. However, the loss of SF3B2 methylation does not seem to affect the integrity of the SF3B complex (Supplementary Fig. 8a, b) or the interaction of SF3B2 with U2 snRNA (Supplementary Fig. 8c). To investigate how SF3B2 R508 methylation might affect RNA splicing, we analyzed a few recently published Cryo-EM structures of the human spliceosome complex[51–54] and found that R508 (highlighted in green) either resides in close proximity to or interacts with (e.g., PDB: 6FF4) the nucleotide located 13 nt upstream of the

**Fig. 5 | PRMT9 regulates pre-mRNA splicing of genes involved in synapse development and function. a** SF3B2 is the primary substrate of PRMT9 in mouse hippocampus and it is highly methylated in vivo. PRMT9 only produces [3]H-labeled SF3B2 methylation signal in the KO, but not WT cells (black triangle). Samples were visualized by Ponceau staining (n = 3). **b** Violin plot demonstration of alternative splicing (AS) events identified in *Prmt9 KO* hippocampus. The middle line of the boxplot represents median value. The low and high ends of the box represent the 25% and 75% quantiles, respectively. The two whiskers extend to 1.5 times the interquartile range. The number of significant events within each category are indicated within parentheses along x-axis labels. WT: n = 3; KO: n = 3. **c** Scatter plot of gene expression levels in WT and *Prmt9 KO* hippocampus samples. Genes with significant changes in alternative splicing are depicted in yellow. Genes selected for RT-PCR validation of AS are highlighted in boxes. **d** Pathway enrichment analysis of alternatively spliced genes. Pathways were grouped by database resource origins (BioPlanet pathway or WikiPathway). The length of bars depicts the Benjamini-Hochberg adjusted *p* values calculated from a one-sided hypergeometric test. Odds ratio of the enrichment is indicated by bar opacity. **e** RT-PCR validation of alternatively spliced genes upon *Prmt9* KO. RNA-seq read coverage across individual alternatively spliced exons in WT and *Prmt9* KO samples is illustrated using the Sashimi plots. The number of reads mapped to each splice junction is shown. Percent Spliced In (PSI) values are indicated on the upper-right sides of the plots. RT-PCR was performed to validate selected alternative splicing events. Inclusion versus exclusion ratio was calculated. Data are presented as mean values ± SD. Error bars represent standard deviation calculated from three independent experiments. The splice site highlighted by a red triangle is a cryptic splice site that is not annotated in the reference genome (for the *Celf5* gene). *, $P < 0.05$; **, $P < 0.01$; ***, $P < 0.001$. Source data are provided as a Source Data file.

pre-mRNA branch point adenosine (A) (Fig. 7a). Thus, we hypothesized that R508 methylation by PRMT9 might be involved in the regulation of protein RNA–interaction between SF3B2 and the pre-mRNA anchoring site.

To test this hypothesis and determine the functional significance of R508 methylation in splicing regulation, we focused on a PRMT9-regulated splicing target, *Stag2*, whose function has been linked to brain development and cognition[55–57], and constructed a *Stag2* splicing minigene (Fig. 7b, top). Transfection of this minigene in control and *PRMT9* KO HEK293T cells faithfully recapitulated the splicing pattern in hippocampus (compare Fig. 7b bottom left with Fig. 5e). Importantly, expression of the SF3B2 methylation-deficient mutant (R508K) resulted in similar exon inclusion as PRMT9 KO (Fig. 7b, bottom right), further confirming that the effect of PRMT9 loss on *Stag2* splicing is caused by SF3B2 methylation deficiency. Based on the consensus sequence analysis, *Stag2* is predicted to contain a PRMT9-regulated anchoring site in the upstream intron (Fig. 6e). Thus, we generated mutant *Stag2* minigenes with single nucleotide variations at −12, −13, and −14 nt of the upstream anchoring sequence vicinity of the R508 interaction site (Fig. 7c, left panel). Interestingly, nucleotide variations at −13 nt, but not at −12 nt or −14 nt, exhibited significantly different splicing patterns, with two pyrimidine bases (T or C) strongly promote exon inclusion (Fig. 7c, right panel). Surprisingly, when expressed in PRMT9 KO cells, all these minigenes exhibited high levels of exon inclusion (Fig. 7d), suggesting that R508 methylation deficiency enhances their exon inclusion. Notably, PRMT9 KO, relative to control, has a milder effect on minigenes harboring pyrimidine bases at −13 nt (Fig. 7d), which might explain why PRMT9-regualted alternative exons exhibited more nonconsensus anchoring sequences in comparison to the native exons (Fig. 6e).

To further investigate the molecular basis underlying the alternative splicing caused by PRMT9 KO, we tested if PRMT9-mediated R508 methylation differentially regulates SF3B2 interaction with non-consensus vs. consensus anchoring sequences. Using anchoring sequences derived from the *Stag2* minigenes (Supplementary Fig. 8d, left panel), we performed electrophoretic mobility shift assays (EMSA) to measure their interactions with unmethylated (R508me0) and SDMA-modified (R508me2s) SF3B2 peptides. The results showed that lack of methylation (R508me0) enhances SF3B2 interaction with both anchoring sequences and there is no obvious difference between nonconsensus (upstream, probe A) and consensus (downstream, probe B) anchoring sequences (Supplementary Fig. 8d, right panel). To detect these interactions in vivo, we performed cross-linking immunoprecipitation (CLIP)-qPCR and compared SF3B2 recruitment in control and *Prmt9* KO hippocampus. Upon PRMT9 KO, the interactions between SF3B2 and the nonconsensus, but not the consensus anchoring sites, were significantly enhanced (Fig. 7e), regardless of whether the nonconsensus anchoring site resided in the upstream intron (for the *Stag2* gene) or in the downstream intron (for the *Stxbp5l* and *Grin1* genes). These results suggest that R508 methylation negatively regulates the interaction of SF3B2 with nonconsensus anchoring sites in a context-dependent manner, likely involving additional features of the 3′SS and RBPs in vivo.

## Discussion

Based on the results presented, we propose the following working model: PRMT9-mediated SF3B2 R508me2s regulates RNA splicing through modulating SF3B2–anchoring site interaction and 3′ splice site selection. For exons whose upstream intron is associated with overall weaker 3′ splice site features (e.g., weaker 3′ splice sites, weaker branch point sequences, and nonconsensus anchoring site sequences), the usage of this 3′ splice site would be enhanced by SF3B2 methylation loss (caused by PRMT9 KO or the ID-associated PRMT9 loss-of-function mutation), resulting in more exon inclusion. Vice versa, if the weak 3′ splice site features reside in the downstream intron, SF3B2 methylation loss would enhance the usage of downstream 3′ splice site, thus, causing more exon skipping (Fig. 7f).

In 2015, we identified SF3B2 as an interaction protein and methylation substrate of PRMT9[12,13]. Over the past few years, we have attempted to identify additional PRMT9 substrates using various proteomic approaches, including proximity-dependent biotin identification (BioID)[58] and immunoaffinity purification, but to no avail. Several lines of evidence suggest that SF3B2 is the primary, if not only, substrate of PRMT9: 1) PRMT9 forms a tight protein complex with SF3B2[13]. This high-affinity enzyme–substrate interaction is distinct from other PRMTs, whose interactions with their substrates are often transient and weak; 2) PRMT9 requires a stringent substrate motif for methylation[12] and, in a large peptide library screening, only methylates SF3B2[59]; 3) PRMT9 marginally contributes to total cellular SDMA, whereas knockout of PRMT5, the primary SDMA-forming enzyme, results in an almost complete loss of SDMA[12]; 4) on native total cell-lysate substrates, recombinant PRMT9 only methylates a single band that corresponds to SF3B2 protein in PRMT9 KO lysates, but not WT lysates, strongly suggesting that SF3B2 R508 is the major PRMT9 substrate and that SF3B2 R508 is highly, if not fully, methylated in WT cells (Fig. 5a and Supplementary Fig. 5); and 5) SF3B2 arginine methylation deficiency resembles PRMT9 KO in causing defective synaptic plasticity, cognitive function, and hippocampus RNA splicing (Supplementary Fig. 7), suggesting that the biological function of PRMT9, at least in neurodevelopment, is largely mediated by its activity in methylating SF3B2. While a recent study suggested that MAVS is a substrate for PRMT9[41], we were unable to detect MAVS methylation under our assay conditions that detect strong SF3B2 methylation (Supplementary Fig. 5e). However, it is possible that MAVS is methylated at a level below the detection threshold of our assay. Nevertheless, collective evidence strongly supports that SF3B2 is the primary methylation substrate of PRMT9.

The high stoichiometry of a specific PTM often indicates its housekeeping role in maintaining protein function[60,61]. In the case of SF3B2

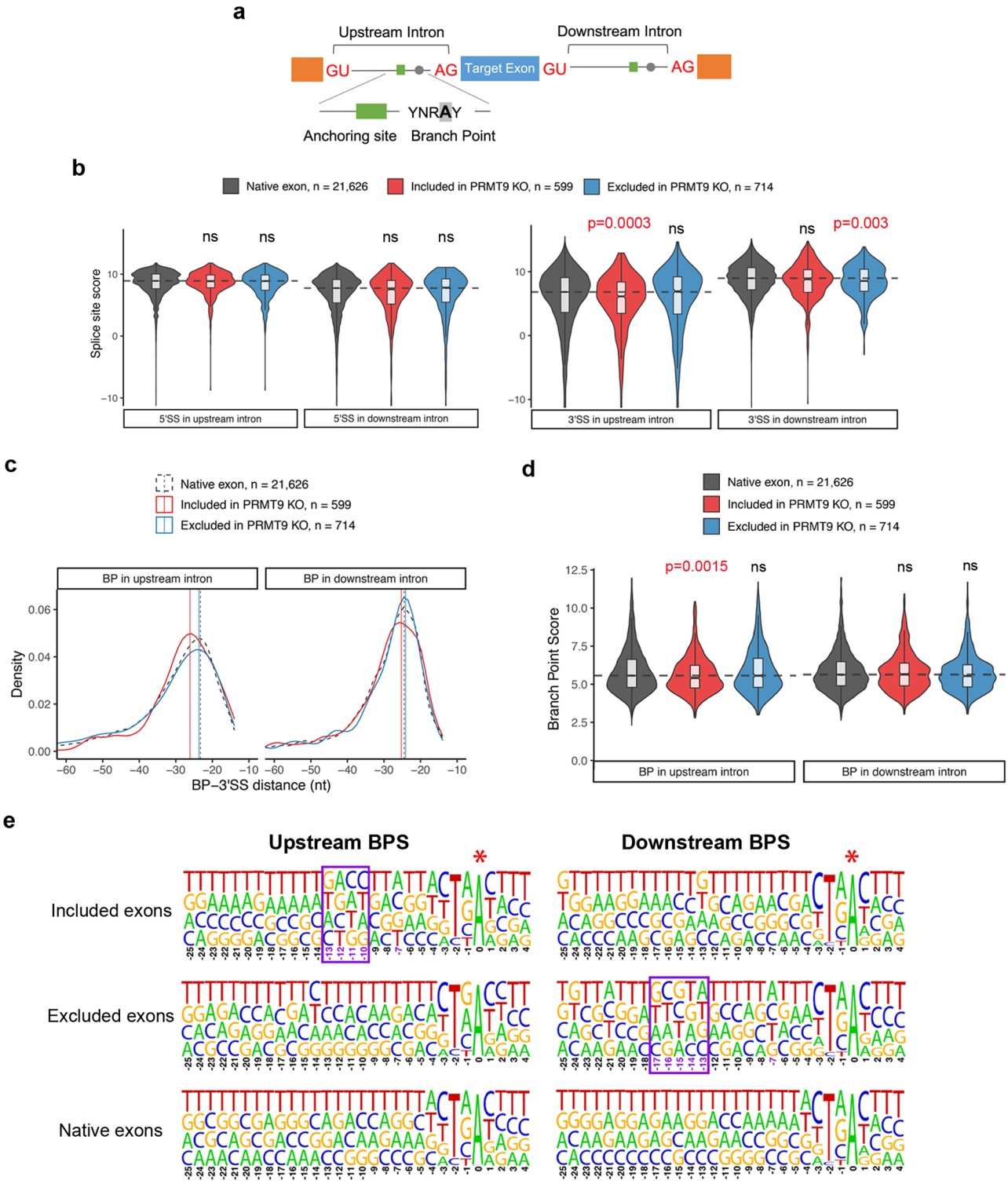

R508me2s, our study suggests that this PTM is particularly involved in the regulation of SF3B2 interactions with a subset of PRMT9-sensitive anchoring sequences that we call nonconsensus anchoring sites. Specifically, the lack of SF3B2 methylation upon PRMT9 KO enhances SF3B2 engagement with the nonconsensus anchoring sites to promote the usage of its corresponding 3′ splice site (Fig. 7e). Although the precise sequence features of these nonconsensus anchoring sites remain to be defined, results from a series of experiments using *Stag2* minigene mutations suggest that the splicing of introns harboring purine bases (A or G) at −13 nt upstream of the branch point might be more likely to be affected by PRMT9 loss (Fig. 7a–d). Note that these

introns also exhibit overall weaker 3′ splice site and branch point sequences (Fig. 6b–d). Thus, a nonconsensus anchoring sequence might not be the only determinant for whether the splicing of a pre-mRNA transcripts will be affected by PRMT9 loss. Although lack of methylation at R508 in general enhances SF3B2 interaction with anchoring sequences (Supplementary Fig. 8d) in vitro, this effect becomes context-dependent in cells (Fig. 7e).

It is intriguing that a single nucleotide variation at the −13 nt of the anchoring site is sufficient to impact *Stag2* minigene splicing (Fig. 7c). Although the extent to which this phenomenon could be generalized remains to be determined, it highlights the critical role of the

**Fig. 6 | PRMT9-regulated splicing targets exhibit nonconsensus anchoring site sequences. a** A schematic illustration of splicing cis-regulatory elements located in the intron sequence of pre-mRNA. Branch point and anchoring site locations are highlighted in gray and green, respectively. **b** Violin plot demonstration of the maximum entropy score of the 5′ and 3′ splice sites in upstream or downstream introns of differentially spliced cassette exons. The middle line of the boxplot represents median value. The low and high ends of the box represent the 25% and 75% quantiles, respectively. The two whiskers extend to 1.5 times the interquartile range. The statistical significance against background native cassette exons was assessed using a two-sided Wilcoxon's rank-sum test. ns, not significant. **c** Density plot of the relative positions of predicted branch point (BP) locations distanced from their corresponding 3′ splice sites. Native exon, n = 21,626; Included in PRMT9 KO, n = 599; Excluded in PRMT9 KO, n = 714. **d** Violin plot demonstration of BP scores from predicted branch point sites. The middle line of the boxplot represents

median value. The low and high ends of the box represent the 25% and 75% quantiles, respectively. The two whiskers extend to 1.5 times the interquartile range. The statistical significance against background native cassette exons was assessed using a two-sided Wilcoxon's rank-sum test. ns, not significant. Native exon, n = 21,626; Included in PRMT9 KO, n = 599; Excluded in PRMT9 KO, n = 714. **e** Sequence logo demonstration of the nucleotide frequency upstream of the predicted branch point sites. The height of the symbols within the stack indicates the observed frequency of the corresponding nucleotide at that position. The 0 point marks the position of the branch point adenosine (marked with an asterisk). Sequences from −25 nt to 4 nt relative to the branch point were shown, which include the reported anchoring site sequence covering 6 to 25 nt upstream of the branch point. The boxes highlight the sequence variations in PRMT9-regulated alternatively spliced exons in comparison to all native exons.

anchoring sequence as an essential cis-regulatory element in splicing regulation. Several core components of the U2 snRNP complex, including SF3B1, SF3B2, SF3B4, SF3A1, SF3A2, and SF3A3, directly interact with the anchoring site[21,22]. Structural analysis revealed that these dense protein–RNA interactions could be involved in constraining the length of the U2/intron duplex to be 16 bps (−13 nt to +2 nt around BP)[51–54]. Importantly, the −13 nt is located at the 5′ end of the U2/intron duplex, where U2 snRNA starts to form stem loop IIa (SLIIa) onward[62,63]. Thus, protein–RNA and RNA–RNA interactions at this location could potentially affect U2 snRNP/intron engagement and RNA splicing.

Alternative splicing regulation is highly pervasive in the central nervous systems[23,64]. Various aspects of brain development and function, including neurogenesis, synaptogenesis, and the homeostasis of neuronal activity, involve alternative splicing regulation[23,24,65]. Specifically, alternative splicing can directly impact glutamate receptors, plasticity, and maturation of cortical circuits[23,66]. Spine size and density are highly correlated with glutamate receptor content and degree of maturation[67–69]. Our observation that CA1 neurons from *Prmt9* cKO mice show decreased spine density and sizes, and that cultured primary cKO neurons exhibit reduced numbers of putative functional synapses further support a critical role of PRMT9/SF3B2-mediated splicing in neurodevelopment. An earlier study comparing the transcriptome changes between mouse fetal and adult cerebral cortex revealed dramatic differences in alternative splicing isoforms during brain development. Surprisingly, more than 30% of the genes that were found to be differentially regulated by alternative splicing did not change their total expression levels[70], suggesting that alternative mRNA isoforms are the major regulatory paradigm for the expression and function of these genes. In support of this, the genetic knockout of NOVA, a neuronal-specific splicing factor that regulates ~7% of brain-specific splicing, in neocortex causes dramatic changes in alternative splicing, but not significant changes in steady-state gene expression levels[42], similar to transcriptome changes upon PRMT9 KO in hippocampus (Fig. 5c). These results suggest that like NOVA, PRMT9 does not have a general role in regulating gene transcription or RNA turnover, but rather acts through alternative splicing to define the hippocampus-specific transcriptome. Notably, PRMT9 also shares a significantly numbers of splicing targets with NOVA (Fig. 5d), including *Grin1* and R*ap1gap* (Fig. 5e and Supplementary Fig. 6e), suggesting that they might function cooperatively to regulate splicing in synapse development, plasticity, and cognitive function.

## Methods
### Cell culture and transfection
HeLa and HEK293 cells obtained from ATCC were cultured in Dulbecco's modified Eagle's medium (DMEM) supplemented with 10% fetal bovine serum (FBS), 100 U/ml penicillin and 100 µg/ml streptomycin at 37 °C in an atmosphere of 5% $CO_2$. Transient transfection was performed using the PolyJet™ In vitro DNA transfection reagent

according to the manufacturer's instructions (SignaGen Laboratory). In all transfections, the total amount of DNA was normalized by using empty control plasmids. For small-interfering RNA (siRNA) transfection, we used Lipofectamine 2000 reagent according to the manufacturer's instructions (Invitrogen). Dharmacon™ pooled gene-specific and negative control siRNA were purchased from Horizon Discovery. Cells were prepared for immunoblotting 72 h after siRNA transfection. Mouse embryonic stem cells (mESCs) were cultured in feeder-free 0.1% gelatin-coated plates. The culture medium contained DMEM supplemented with 15% stem cell qualified FBS, 0.1 mM non-essential amino acids solution (NEAA), 100 U/ml penicillin, 100 µg/ml streptomycin, 0.1 mM β-mercaptoethanol (BME), and 1000 U/ml mouse leukemia inhibitory factor (mLIF).

### Plasmids
Flag-PRMT9, GFP-PRMT9, Flag-SF3B2, Flag-SF3B2 (R508K) GST-SF3B2 (401-550) and HA-Ubiquitin have been described previously[13,71]. Flag-PRMT9 (G189R) and Flag-PRMT9 4A mutant constructs were generated by site-directed mutagenesis according to the QuikChange XL Site-Directed Mutagenesis kit (Agilent Technologies). Flag-UBE3C (WT) and Flag-UBE3C (C1051S) were kindly provided by Dr. Ruey-Hwa Chen (National Taiwan University). They were subcloned into the pEGFP-C1 vector to generate GFP-UBE3C (WT) and GFP-UBE3C (C1051S). pSpCas9(BB)−2A-Puro (PX459) V2.0 was a gift from Feng Zhang (Addgene plasmid # 62988). Human Mitochondrial antiviral-signaling protein (MAVS) was amplified from reverse transcribed cDNA from HEK293T cells and cloned into the pGEX-4T-1 and p3xFLAG-CMV-7.1 vectors.

To construct the Stag2 splicing minigene, we first performed overlapping extension PCR to obtain genomic sequences that contain exon 30−32, as well as the intron sequences in between. The length of the intron between exon 31 and exon 32 is ~3 kb, so we only included 150 bp of the 5′ splice site and 150 bp of the 3′ splice site, which is sufficient to maintain essential splicing regulatory elements. The anchoring site mutated minigenes were generated by site-directed mutagenesis (Agilent Technologies). All plasmid constructions were confirmed by DNA sanger sequencing.

### Antibodies
The mouse monoclonal PRMT9 antibody and SF3B2 R508me2s antibody were described previously[13]. Anti-Flag (F3165-1MG), anti-Flag M2 magnetic beads (M8823-5ML), and anti-ACTIN (A2228-200UL) were purchased from Sigma. Anti-Ubiquitin (#3933), anti-HA (#3724), anti-GST (#2622), anti-MCL1 (#94296), and anti-Tubulin (#2144) antibodies were purchased from Cell Signaling Technology. Anti-SF3B2 (NB100-79848) and anti-SF3B4 (NBP2-20327) antibodies were purchased from Novus Biologicals. Anti-VCP (A300-589A) and anti-UBE3C (A304-122A) antibodies were purchased from Bethyl Laboratory. Anti-SF3B1 (SC-514655) was purchased from Santa Cruz Biotechnology. Anti-SF3A1 (PA5-51439) was purchased from Thermo Fisher. Anti-PSD95

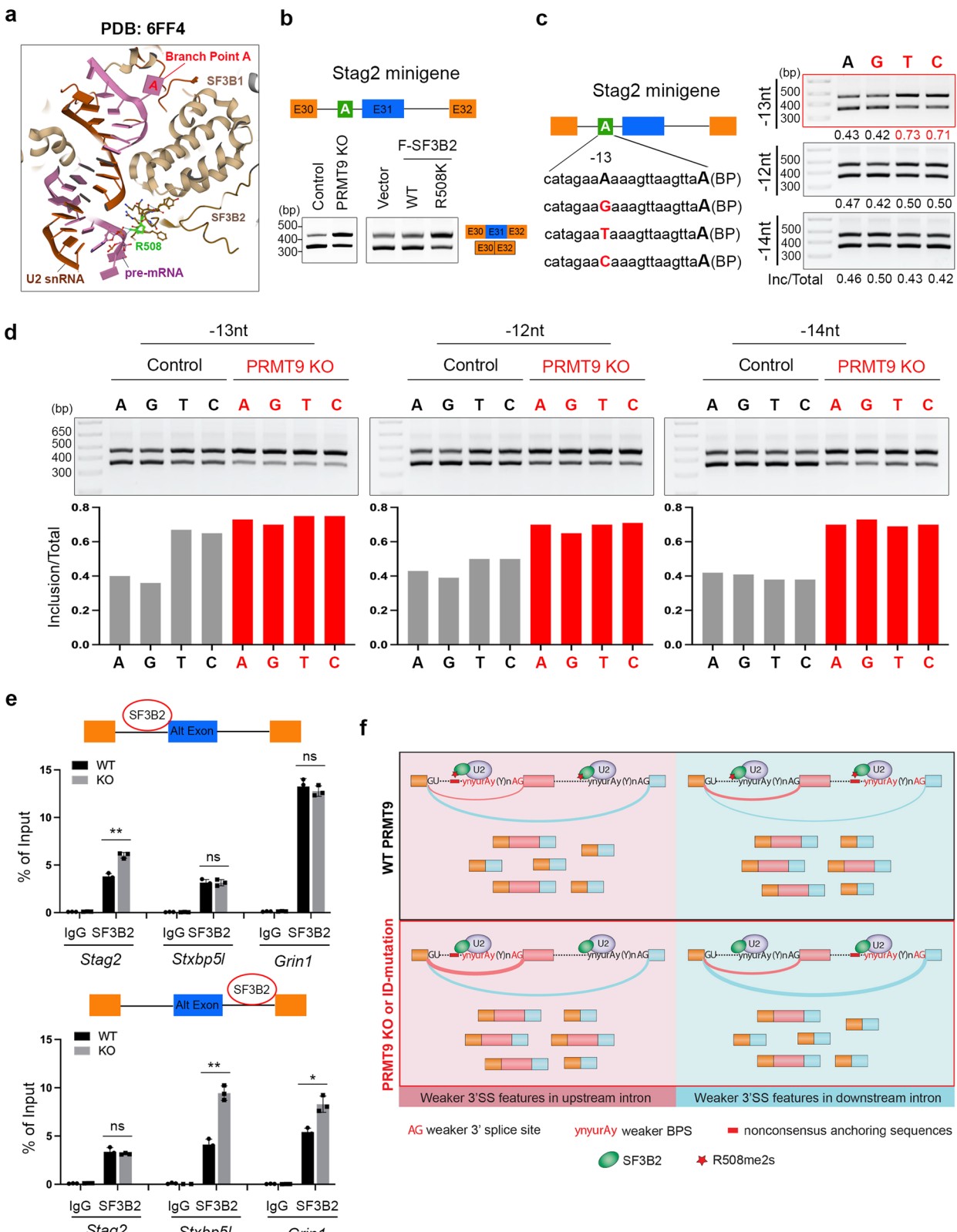

(MABN68), anti-SYN (AB1543), anti-GluN1 (MAB1586), and anti-GluA1 (AB5849) were purchased from Millipore. The rabbit polyclonal PRMT9 antibody targeting mouse PRMT9 protein was generated using a mouse PRMT9 peptide (a.a. 794–a.a. 812), CLDDEVRLDTSGEASHWKQ (YenZym Antibodies, LLC). The specificity of this antibody was characterized by using Prmt9 knockout cell lines.

**Protein sequence alignment using ClustalW**

The parameters for the alignment using ClustalW were set as the following: Gap Penalty: 10, Gap Length Penalty: 0.2, Delay Divergent Seqs (%) 30, Protein Weight Matrix: Gonnet Series for multiple alignment parameters. For pairwise alignment, Gap Penalty: 10, Gap Length 0.1, Protein Weight Matrix: Gonnet 250.

**Fig. 7 | PRMT9-mediated SF3B2 R508me2s regulates splicing through the pre-mRNA anchoring site. a** Cryo-EM structure of the human spliceosome complex (PDB: 6FF4). Branch point on the pre-mRNA is shown in red. SF3B2 R508 is highlighted in green. **b** A schematic illustration of the *Stag2* splicing minigene. The alternative exon 31 is labeled in blue. The anchoring site is highlighted in green (upper panel). RT-PCR was performed to detect the alternative splicing products in control and *PRMT9* KO HEK293 cells, as well as in cells expressing WT and R508K mutant SF3B2 (lower panel). n = 3. **c** Single nucleotide variations at the anchoring site differentially regulate *Stag2* minigene splicing. WT and mutant Stag2 splicing minigenes that contain single nucleotide mutations at −12, −13, and −14 nt upstream of the BP were compared by RT-PCR. The relative ratios of inclusion vs total (Inc/Total) were quantified using ImageJ software (n = 3). **d** *PRMT9* KO promotes exon

inclusion and renders splicing to be insensitive to −13nt nucleotide variations. The splicing patterns of *Stag2* minigenes containing WT or single nucleotide mutations at −12, −13, and −14 nt upstream of the BP were detected in control and *PRMT9* KO HEK 293 T cells by RT-PCR (n = 3). **e** PRMT9 negatively regulates SF3B2 interaction with nonconsensus anchoring sites. CLIP-qPCR was performed with the hippocampus tissues from WT and *Prmt9* KO mice using the SF3B2 antibody. The amount of SF3B2-bound RNA was quantified by qPCR. Data are presented as mean ± SD (n = 3). *, $p < 0.05$; **, $p < 0.01$, ns, not significant. **f** Proposed working model: PRMT9-mediated SF3B2 R508me2s regulates RNA splicing through modulating SF3B2–anchoring site interaction and 3′ splice site selection. Source data are provided as a Source Data file.

## Protein structure and modeling
PRMT7 (PDB ID: 4C4A) and PRMT9 structures (PDB ID: 6PDM) downloaded from PDB were visualized, aligned and G189R mutagenesis performed using PyMOL.

## Generation of CRISPR knockout cell lines
Prmt9 knockout mESCs were generated by transfecting gBlock gene fragments (Integrated DNA Technologies) containing a guide RNA sequence that targeted the mouse Prmt9 gene (5′- GAACCGTTCTGG ATATCGGC −3′). PRMT9 knockout HeLa and HEK293T cells were generated by transfecting the hSpCas9-2A-Puro vector (PX459) containing different guide RNA sequences targeting three different genomic regions of human PRMT9 gene (sgRNA1: 5′- AGTCGGATAC AAGACTTACT −3′; sgRNA2: 5′- GTTTGCTGCCACGACATCAC −3′; and sgRNA3: 5′- GAAAAGATGAGTCGAGTTCC −3′). All the knockout cell lines were derived from single-cell clones and confirmed by western blot analysis.

## In vitro methylation assay
In vitro methylation reactions were carried out in 30 μl of phosphate-buffered saline (PBS) (pH = 7.4.) containing 0.5–1.0 μg of substrate, 3 μg of recombinant enzymes, and 0.42 μM S-adenosyl-l-[methyl-$^3$H] methionine (79 Ci/mmol from a 7.5 μM stock solution; PerkinElmer Life Sciences). The reactions were incubated at 30 °C for 1 h and then separated on SDS-PAGE, transferred to a PVDF membrane, treated with En3Hance™ (PerkinElmer Life Sciences), and exposed to film for 1–3 days at −80 °C.

In vitro methylation with total cell lysates as substrates was performed as described above except that total cell lysates from HeLa cells, mESCs, E16 mouse embryo, and various mouse tissues were used as the substrates. Tissue and cell lysates were prepared with a mild lysis buffer (50 mM Tris-HCl [pH 7.5], 150 mM NaCl, 0.1% NP-40, 15 mM MgCl$_2$). The lysate was sonicated to maximize the protein extraction.

## Western blot analysis
Cells were lysed in radioimmunoprecipitation assay (RIPA) buffer (50 mM Tris [pH 8.0], 150 mM NaCl, 1% Triton X-100, 0.5% sodium deoxycholate, 0.1% SDS, 2 mM EDTA, and protease inhibitors). For immunoblotting, an equal amount of each sample was resolved by sodium dodecyl sulfate polyacrylamide gel electrophoresis (SDS-PAGE) and transferred to polyvinylidene difluoride (PVDF) western membrane. Following blocking with 5% non-fatty milk in phosphate-buffered saline with Tween 20 (PBS-T) buffer, membranes were incubated with the indicated primary antibodies overnight at 4 °C. The HRP-conjugated secondary antibodies were used against respective primary antibodies. The membranes were visualized by chemiluminescence by using X-ray films.

## Immunoprecipitation (Co-IP)
Cells from 10 cm dishes were washed with ice-cold PBS and lysed with 1 ml of co-IP buffer (50 mM Tris-HCl [pH 7.5], 150 mM NaCl, 5 mM EDTA, 15 mM MgCl$_2$, 0.1% NP-40 and protease inhibitor cocktail). Cell

lysates were briefly sonicated, and insoluble materials were cleared by centrifugation at maximum speed for 15 min at 4 °C. Total cell lysates were incubated with 2 μg of immunoprecipitation antibody overnight with gentle rotation. After incubation with Protein A/G agarose beads at 4 °C for 2 h, the immune complex was precipitated by centrifugation and washed with Co-IP buffer three times. After the final wash, Protein A/G beads were resuspended with 40 μl 2× protein loading buffer and analyzed by western blot.

## Immunofluorescence (IF)
Cells were grown on coverslips in 24-well plates, fixed with 4% formaldehyde for 15 min, and permeabilized with PBS containing 0.3% Triton X-100. After blocking with 5% bovine serum in PBS for 1 h at room temperature, the coverslips were incubated with primary antibody in blocking buffer overnight at 4 °C. After washing three times with PBS-T buffer, the coverslips were incubated with Alexa Fluro 488-, or 555-conjugated secondary antibody and nuclei were identified by staining the coverslips with 4′,6-diamidino-2-phenylindole (DAPI). Images were acquired by using a ZEISS Axio Observer and processed using the ZEN Blue software (Zeiss).

## Protein half-life assay
Cells were transfected with the indicated plasmids for 48 h prior to treatment. To inhibit protein synthesis, cycloheximide (CHX) at a final concentration of 50 ug/ml was added to cell culture medium for the indicated time periods. The total cell lysates were prepared using the RIPA buffer after treatment. The expression levels of the indicated proteins were detected by western blot analysis. Quantification of the western blot signal was performed by using the ImageJ software.

## Protein purification
For the purification of GST-tagged proteins, Escherichia coli strain BL21 (DE3) was transformed with the indicated plasmid, and a single colony was picked and cultured in 5 ml LB Broth with 100 μg/ml ampicillin overnight. 45 ml fresh LB Broth with 100 μg/ml ampicillin was added the next day. At OD = 0.6, protein expression was induced with 1 mM IPTG at 30 °C for 4 h or 16 °C overnight. The cells were sonicated in PBS on ice and clarified by centrifugation. The lysates were subsequently incubated with Glutathione Sepharose 4B resin (GE Healthcare Life Sciences) overnight at 4 °C. After washing three times with PBS, the GST-tagged proteins were eluted with 10 mg/ml reduced L-Glutathione in elution buffer (100 mM Tris−HCl [pH 7.4], with 150 mM NaCl).

For the purification of Flag-tagged recombinant proteins, HEK293T cells were transfected with the indicated plasmids for 48 h and lysed in M2 binding buffer (20 mM Tris−HCl [pH 7.5], 150 mM NaCl, 1% NP-40, 2 mM EDTA and protease inhibitors) at 4 °C for 30 min. The lysates were briefly sonicated on ice and clarified by centrifugation. The lysates were subsequently incubated with anti-Flag M2 magnetic beads overnight at 4 °C. After washing three times with M2 wash buffer (20 mM Tris−HCl [pH 7.5], 500 mM NaCl, 1% NP-40, 2 mM EDTA, and protease inhibitors), Flag-tagged proteins were eluted with

200 µg/ml 3×Flag peptide in TBS buffer (50 mM Tris–HCl [pH 7.4] with 150 mM NaCl).

## Liquid chromatography–mass spectrometry (LC/MS)

Protein complexes purified by Flag-tagged WT and G189R-mutant PRMT9 were resolved on a 4–20% Mini-PROTEAN® TGX™ Precast Protein Gel (Bio-Rad, # 4561094) and stained with Coomassie brilliant blue (Thermo Fisher, #20279). The protein band was excised and destained, followed by in-gel digestion using Trypsin/Lys-C Mix (Promega, cat. no. V5073), according to the manufacturer's instructions. After overnight digestion, the peptides were extracted by adding 50% ACN/0.1% TFA solution, 60% ACN/0.1% TFA solution, and 80% ACN/0.1% TFA solution to the gel pieces three times. The combined peptide extracts were evaporated using a Savant SpeedVac SVC 100H Centrifugal Evaporator. The peptides were dissolved in 1% formic acid (Fisher Chemical, #A11750) and analyzed by reversed-phase LC/MS. The mass spectrometric analysis was carried out using a Thermo Scientific Orbitrap Fusion Mass Spectrometer equipped with an Easy Spray source and an Easy-nLC1000 system. The raw spectra files were searched using both Proteome Discoverer Software with Sequest (Version 2.0) and the Mascot algorithm (Mascot 2.5.1). The identified interacting proteins were listed in Supplementary Data 1.

## Quantitative reverse transcription PCR (RT-qPCR)

Total RNAs from cell lines or hippocampus tissue were isolated using TRIzol reagent (Cat# 15596018, Invitrogen) following the manufacturer's instructions. Complementary DNA (cDNA) was prepared using the High-Capacity cDNA Reverse Transcription Kit (Cat# 4368813, Applied Biosystems). Quantitative PCR was performed using Power SYBR™ Green PCR Master Mix (Cat# 4368706, Applied Biosystems) and the CFX Opus 96 Real-Time PCR System (Bio-Rad Laboratories). Data analysis was performed using the Bio-Rad CFX Manager 3.1. The experimental cycle threshold (Ct) was calibrated against the ACTIN control. All amplifications were performed in triplicate.

## In vivo ubiquitination assay

293 T cells were transfected co-transfected Flag-PRMT9 WT or Flag-PRMT9 G189R as substrates and GFP-UBE3C WT or GFP-UBE3C C1951S as E3 ligases, with or without HA-ubiquitin. 48 h after transfection, cells were harvested and lysed in RIPA buffer supplemented with protease inhibitors. Flag-tagged PRMT9 or G189R mutants were purified by M2 beads and detected by Immunoblotting.

## In vitro ubiquitination assay

In vitro ubiquitination assay was performed in a 20 µl reaction mixture containing 25 mM HEPES [pH 7.5], 200 mM NaCl, 5 mM MgCl$_2$, 2.5 mM ATP, 50 µM ubiquitin, 200 nM E1, 500 nM E2, and 2 µM GST-UBE3C or GST-UBE3C C1051S (E3) (HECT domain, purified from E. coli strain BL21 (DE3)), together with 500 ng Flag-PRMT9 WT or Flag-PRMT9 G189R at 37 °C for 90 min. The reaction was stopped by the addition of SDS loading buffer, and the mixture was resolved and analyzed by Immunoblotting.

## Electrophoretic mobility shift assays (EMSA)

The 5′ 6-FAM labeled ssRNA oligonucleotide from upstream and downstream anchoring sites of the *Stag2* Exon 31 (upstream probe A: 5′- rCrArUrArGrArArArArArArGrUrUrArGrUrUrUrArA-3′; downstream probe B: 5′ rUrArArUrArGrUrCrArUrGrCrCrUrUrGrGrUrUrCrA-3′) were purified by PAGE gel and incubated at a final concentration of 1 µM with increasing amounts (1.4 µM, 14 µM, and 140 µM) of either unmodified (R508me0: RNSVPVPRHWCFKRKYLQGKRGIEKPP) or R508 methylated (R508me2s: RNSVPVPRHWCFKR$^{me2s}$KYLQGKRGIEKPP) SF3B2 peptides at room temperature in 10 µl reaction buffer containing 20 mM Tris-HCl [pH 8.0], 150 mM NaCl, 1.5 mM MgCl$_2$, 0.5 mM EDTA, and 100 U/mL RNasin for 20 min. The reactions were then resolved on 6 % native

acrylamide gels (37.5:1 acrylamide:bis-acrylamide) in 0.5× TBE buffer. The mobility shift of oligonucleotides was detected using Bio-Rad ChemiDoc Imaging System.

## Cross-linking immunoprecipitation and qPCR (CLIP-qPCR)

Hippocampus region from WT or Prmt9 KO mice were homogenized with 27 G needles and then cross-linked under 400 J/cm² 365 nm UV. The crosslinked hippocampus samples were then lysed in lysis buffer (150 mM KCl, 25 mM Tris [pH 7.4], 5 mM EDTA, 0.5 mM DTT, 0.5% NP40, and 100 U/mL RNase inhibitor) for 1 h at 4 °C, followed by sonication using the Bioruptor ® Pico sonication device (Diagenode) at a setting of 30″ on/ 30″ off for 10 cycles. The sonicated cell lysates were clarified by centrifugation at maximum speed for 15 min at 4 °C. For immunoprecipitation, the lysates were incubated with 1 µg of SF3B2 antibody or control IgG for 4 h with gentle rotation at 4 °C, followed by incubation with protein A Dynabeads (Invitrogen) for 2 h at 4 °C. The immune complex was washed with lysis buffer three times. After the final washing, the beads were incubated in 100 µl of elution buffer (50 mM Tris [pH 7.5], 75 mM NaCl, 6.5 mM EDTA, 1% SDS and 0.2 mg/ml Proteinase K) at 50 °C for 1 hr. RNA was then extracted by using Phenol-chloroform, and reverse transcribed into cDNA using the High-Capacity cDNA Reverse Transcription Kit. Equal volumes of RNA from each sample were used for the reverse transcription. The relative amount of immunoprecipitated RNA was quantified by qPCR analysis. All amplifications were performed in triplicate. The primers for qPCR are listed in Supplementary Data 2.

## RNA immunoprecipitation (RIP)-qPCR

Control and PRMT9 KO HEK293T cells were crosslinked with 1% formaldehyde for 10 min at room temperature. Crosslinking was stopped by the addition of glycine to a final concentration of 0.25 M for 5 min. Cells were washed three times with ice-cold PBS and lysed in RIPA buffer (50 mM Tris [pH 7.5], 150 mM NaCl, 1% Triton X-100, 0.5% sodium deoxycholate, 0.1% SDS, and 2 mM EDTA, supplemented with protease inhibitors and RNase inhibitor) for 1 h at 4 °C, followed by sonication using the Bioruptor ® Pico sonication device (Diagenode) at a setting of 30″ on/ 30″ off for 10 cycles. Insoluble materials were cleared by centrifugation at maximum speed for 15 min at 4 °C. Cell lysates were incubated with 1 µg of SF3B2 antibody or control IgG overnight at 4 °C with gentle rotation. After incubation with Dynabeads beads for 2 h at 4 °C, the immune complex was washed with washing buffer I (50 mM Tris–HCl [pH 7.5], 1 M NaCl, 1% NP-40, and 1% sodium deoxycholate) and then washed three times for 5 min with washing buffer II (50 mM Tris–HCl [pH 7.5], 1 M NaCl, 1% NP-40, 1% sodium deoxycholate, and 1 M urea). After the final washing, the beads were incubated in 100 µl of elution buffer (100 mM Tris–HCl [pH 8.0], 200 mM NaCl, 10 mM EDTA, 1% SDS, and 0.2 mg/ml Proteinase K) for 1 h at 42 °C, followed by 1 h at 65 °C. RNA was then extracted by using Phenol-chloroform, and reverse transcribed into cDNA using the High-Capacity cDNA Reverse Transcription Kit. The relative amount of immunoprecipitated RNA was quantified by qPCR analysis. All amplifications were performed in triplicate. The primers for qPCR are listed in Supplementary Data 2.

## Prmt9$^{Flox/Flox}$ and Sf3b2$^{R491K}$ mouse models

The *Prmt9* conditional knockout (cKO) mouse model (C57BL/6) was generated using CRISPR/Cas9 technology (Applied StemCell). Briefly, validated gRNA and single-stranded oligodeoxylnucleotide (ssODN) donor, along with the Cas9 protein were used for microinjection to insert two LoxP sequences (5′- ATAACTTCGTATAGCATACATTA-TACGAAGTTAT −3′) into intron 4 and intron 5 of the mouse Prmt9 gene. PCR and Sanger sequencing were performed to confirm the knock-in of the sequence in F0 founders and germline transmission in F1 mice. Subsequent genotyping was performed using PCR. Genotype primers are listed in Supplementary Data 2.

The *Sf3b2*[R491K] point mutant mouse model (C57BL/6) was generated by CRISPR/Cas-mediated genome engineering (Cyagen). R491 (R508 in human) is located on exon 12 of the mouse Sf3b2 locus. The p.R491K (CGC to AAG) in donor oligo was introduced into exon 12 by homology-directed repair. Two synonymous mutations p.C488= (TGT to TGC) and p.K490= (AAG to AAA) were introduced to prevent the binding and re-cutting of the sequence by gRNA after homology-directed repair. Validated gRNA, ssODN donor, and the Cas9 protein were co-microinjected into fertilized eggs of C57BL/6 mice. The resulting pups were genotyped by PCR followed by Sanger sequencing. The genotype primers are listed in Supplementary Data 2.

Emx1-Cre (#005628) and CMV-Cre (#006054) mice (C57BL/6J) were purchased from the Jackson Laboratory. All data generated using mice as listed in this manuscript are approved by the Institutional Animal Care and Use Committee of the City of Hope (Protocol number 17070) and the University of Arizona (protocol number 13-478).

### Body composition analysis
Lean and fat mass content were evaluated by quantitative nuclear magnetic resonance using a Bruker Minispec mq60 (Bruker) on 8-week-old mice.

### Morris water maze
A circular pool was placed in a room with prominent visual cues and filled with room temperature (~24 °C) water (made opaque) to a depth of 30 cm. The pool was placed in a room with prominent ambient visual cues. The circular arena was divided into four equal quadrants with defined hidden platform zones. Mice were placed in one of four starting locations in the pool and allowed to swim until they found a submerged hidden platform ($7.5 \times 7.5$ cm size). A video camera was used to record moving tracks at 15 frames per second (fps) using Ethovision XT (Noldus Information Technology, Leesburg, VA). Training phases included four trials per day for eight consecutive days with an inter-trial interval of one hour. A probe trial was conducted on Day 8, right after the training. During probe trial, the submerged platform was removed, and mice were allowed to freely swim in search of the platform. Time spent in the target quadrant, and the number of crosses over original platform location was quantified. A reverse learning test was conducted on Days 9–11, during which mice were trained to relearn a new platform location that was moved to the opposite quadrant. Another probe test was conducted following the completion of the reverse learning training session to assess cognitive flexibility. Only mice that met all the performance criteria through the acquisition/probe trial/reverse learning were included in analyses.

### Fear conditioning
The Pavlovian fear conditioning learning test was conducted essentially the same as we recently described[72]. Mice were tested in sound-attenuating cabinets, in which the visual contextual cues (enclosure's color and texture) could be readily changed. The unconditioned and conditioned stimulus (foot shock and audio cue) were programmed, and videos were recorded using AnyMaze video tracking system (Stoelting, Wood Dale, IL). Locomotor activity was recorded at 25 fps by a digital camera. Mouse immobility was considered freezing if it lasted at least 1 s. Fear conditioning training was conducted with the grid floor, during which five tone−shock/CS−US pairings consisted of a 30-s tone that co-ended with a 1-s (0.5 mA) foot shock. The CS−US pairing was delivered with a 150 s inter-trial interval for the five training sessions. A contextual recall memory test was performed on the second day, during which mice were introduced to the same context for 300 s, and time spent in freezing was quantified.

### Open field test
The open field test was used to assess basal locomotor activity and reactivity to a novel environment. Mice were introduced to an open field ($45 \times 45$ cm$^2$) arena with a 40 cm wall and were habituated for 5 min, followed by 10 min of video recording using Ethovision video-tracking system. Total distance traveled, and the percent of time spent in the center ($22.5 \times 22.5$ cm$^2$) were quantified.

### Elevated plus maze (EPM) test
The EPM test was conducted as previously reported[72] to evaluate anxiety in mice by using their innate preference for dark and enclosed spaces. The EPM apparatus was 4 feet from the ground and consisted of four runways (9 in. × 2.5 in. rectangle) arranged perpendicularly, two of which were enclosed with 5 in. high walls, and two of which were open. The amount of time spent in the closed vs. open arms of the maze was recorded and measured in a 5-min session using Ethovision.

### Hippocampal brain slice preparation
Mice were euthanized with 3–5% isoflurane, and brains were collected from young adult mice of the desired genotypes. To enhance brain slice viability, intra-cardiac perfusion of ice-cold choline solution (110 mM choline chloride, 25 mM NaHCO$_3$, 2.5 mM KCl, 1.25 mM NaH$_2$PO$_4$, 0.5 mM CaCl$_2$, 7 mM MgSO$_4$, 25 mM D-glucose, 11.6 mM sodium ascorbate, and 3.1 mM sodium pyruvate, saturated with 95% O$_2$/5% CO$_2$) was performed. Mice were decapitated and brains were quickly dissected out. Horizontal slices (300 μm thick) at the hippocampus level were made in ice-cold choline solution using a vibratome (VT-1200S, Leica). The hippocampus from each hemisphere was dissected out and kept in artificial cerebrospinal fluid (ACSF, contains 126 mM NaCl, 2.5 mM KCl, 26 mM NaHCO$_3$, 2 mM CaCl$_2$, 2 mM MgCl$_2$, 1.25 mM NaH$_2$PO$_4$, and 10 mM d-glucose, saturated with 95% O$_2$/5% CO$_2$) for 30 min at 35 °C, and then maintained at 22 °C RT until recording.

### Analysis of dendritic spine morphology
CA1 neurons in acute hippocampus slices were filled with 0.25% biocytin during whole cell patch clamp recording, further fixed in 4% PFA overnight. Neuron morphology as revealed by avidin-Alexa 488 (Invitrogen). Z-stacks of spines from apical secondary dendrites (200–450 μm away from the soma) were collected using a 63× objective (Plan-Apochromat, NA 1.4). Spine images were acquired using a $512 \times 512$ pixels frame with 4× digital zoom and 0.2 μm Z step size. Imaris software (V8.02, Bitplane, South Windsor, CT) was used to measure spine head diameter, length, and density.

### Culturing primary embryonic hippocampus neurons
Embryonic hippocampal neuron cultures were prepared from E17.5 time-pregnant Prmt9[fx/fx]:emx1[cre] breeders, similar to that previously described[73]. The hippocampi were dissected out from the embryonic brains and digested with papain. Tails from the embryos were collected for genotyping. The hippocampi were then dissociated into individual cells and plated into 24-well plates with poly-D-lysine-coated glass coverslips at a density of 15,000 cells/cm². Neurons were cultured in Neurobasal medium supplemented with 2% B27 (Life Technologies), and fixed with 4% PFC at DIC 25−28, and processed for immunocytochemistry double labeling with antibody pairs (Synapsin I/PSD95; GluA1/GluN1). Neurons were imaged on a confocal microscope (Zeiss, LSM710) with a 63× oil immersion objective and 2× digital zoom. Imaris (Oxford Instruments) was used to quantify the puncta density and colocalization.

### Field potential recording, LTP and LTD
Hippocampal slices were transferred to an interface chamber (Automate Scientific, Berkeley, CA), and superfused with ACSF. Field excitatory postsynaptic potentials (fEPSPs) were recorded in the CA1 stratum radiatum layer using a glass patch electrode at room temperature to facilitate long-term slice viability. The patch electrode had an electrical resistance of 1–2 MΩ at 1 kHz when

filled with ACSF. fEPSPs were evoked by a tungsten stimulating electrode (FHC Inc, Bowdoin, ME) that was placed on the Schaffer collateral inputs ~200 μm away from the recording site. Biphasic stimulus (graded levels ranging from 10 to 250 μA, 100 μs duration, 0.05 Hz) was generated using a Digidata 1440A interface (Molecular Devices, San Jose, CA) and pClamp 10.2 software. The stimulus was delivered through an optic isolator (Iso-flex, A.M.P.I). fEPSP signals were amplified using a differential amplifier (model 1800, A–M Systems, Carlsborg, WA), low-pass filtered at 2 kHz and digitized at 10 kHz.

A stimulus−fEPSP input−output curve was first obtained by measuring fEPSP slope (first 1-ms response after fiber volley) as a function of the fiber volley amplitude. A stimulus intensity that produced a ~40–50% maximum fEPSP amplitude was then adopted and kept constant throughout the experiments. After establishing a stable baseline response of stimulus-evoked fEPSPs for at least 10 min, an LTP induction stimulation protocol was applied. It is well-established that the synaptic stimulation pattern is critical for LTP/LTD induction and maintenance[74]. For LTP induction, we adopted a tetanus stimulation protocol that consisted of two, 1-s trains of stimulation at 100 Hz[75]. LTD was induced by low-frequency stimulation, which consists of 1 Hz paired pulses (with 50 ms inter-pulse interval) for a 15-min duration (i.e., a total of 900 paired stimuli). fEPSP was continuously recorded for 60 min post LTP/LTD induction.

### RNA-seq
RNA samples were extracted from hippocampus tissue of 2-week-old WT or Prmt9 whole body KO mice (n = 3 for each group). The quality of RNA samples was ensured by calculation of RNA integrity number as well as degradation measurement using the 2200 TapeStation system (Agilent). Poly(A)+ cDNA libraries were subsequently generated using TruSeq stranded mRNA Library Prep Kit. Libraries were sequenced on a NovaSeq 6000 System using S4 flow cell with a paired-end (PE) 2×100 kit at the Translational Genomics Research Institute (TGen).

### Gene expression and alternative splicing analysis
The quality of raw RNA-seq datasets was inspected using FastQC. Reads were aligned to the mouse genome (mm10/GRCm38) by STAR (v2.7.1a)[76] using two-pass mode with Ensembl 97 annotations. Gene expression values were quantified in TPM (Transcripts Per Million) using kallisto (v0.43.1)[77] and subsequently summarized to a gene expression matrix using tximport (v1.6.0, R package)[78]. Differential expression analysis was performed with the count-based tool DeSeq2 (v1.18.1, R package)[79]. Genes with fold change >1.5 and FDR < 0.01 were identified as differentially expressed genes between WT and *Prmt9* KO samples. UCSC genome browser track was utilized to visualize and confirm the change of Prmt9 expression in WT and *Prmt9* KO samples.

Alternative splicing (AS) events detection, quantification and differential splicing analysis were conducted using rMATS-turbo (v4.1.0)[28,30]. Five types of AS events were detected, including exon skipping, alternative 5′ or 3′ splice sites, mutual exclusive exons, and intron retention. Novel splice site detection features of rMATS-turbo were turned on to identify alterations in both annotated and cryptic splicing events. Exon inclusion levels were calculated as Percent Spliced In (PSI) values between 0 and 1, which is the ratio of reads supporting the inclusion isoform to total reads. To enhance the robustness and reliability of the analysis, events with low read support (75 percentile of read count <10 in either group) or extreme PSI value (average PSI value < 0.05 or >0.95 in both groups) were excluded from downstream analysis. Differentially spliced events were further defined by the cut-offs of FDR (≤0.01) and PSI value differences (≥0.05). Virtualization of selected differential splicing events was achieved by rmats2sashimiplot software.

### Gene set enrichment analysis
Genes with differential splicing were tested for enrichment in both Gene Ontology (GO) terms and biological pathways (BioPlanet[80], using Elsevier and WikiPathway[81]), which were retrieved from Enrichr libraries (https://maayanlab.cloud/Enrichr/#libraries)[82]. To eliminate the bias resulting from gene expression on differential splicing analysis, we used a customized background gene list by excluding genes expressed at low levels (DeSeq2 base Mean value < 5) instead of using all genes in the mouse genome. Genes with significant splicing alterations in the top 2 categories (SE and A3SS) were selected as the foreground gene list. The significance of enrichment was then evaluated by a hypergeometric test, and adjusted p values were calculated from the Benjamini-Hochberg procedure.

### Synaptic Gene Ontologies (SynGO) analysis
For further systematic annotation of synaptic genes and ontology of synaptic processes, we performed SynoGO analysis[83]. Specifically, the "brain expressed" gene set downloaded from the SynGO database was selected as background. This gene set contains 18,035 genes, among which 1225 of them were annotated as synapse genes.

### Ras/Rho/PSD95 network analysis
Protein nodes of the Ras/Rho/PSD95 network were curated from two resources[31,43]. Protein-protein interaction edges were collected from the STRING (v11.5) database, with active interaction sources extracted from experimental data (BIND, DIP, GRID, HPRD, IntAct, MINT, and PID) or databases (e.g., Biocarta, BioCyc, GO, KEGG, and Reactome).

### Sequence feature analysis for differential exon skipping events
Comparisons were performed between differentially spliced cassette exons (exon skipping events) and a transcriptome-wide background. The background cassette exon set, or native exons, were defined as exons that were alternatively spliced under normal conditions (0.05 <mean PSI < 0.95 in WT samples).

*Splice site strength*. Splice site sequences were extracted from 5′ splice sites and 3′ splice sites in both upstream and downstream introns. Splice site strengths were calculated using MaxEntScan[84]. The statistical significance of splice site strength differences between differentially spliced cassette exons and native background cassette exons was assessed using Wilcoxon's rank-sum test.

*Branch point prediction and comparison*. Branch point prediction was performed using BPP software[50] in both upstream introns and downstream introns. The software reports both the specific position of a predicted branch point relative to the corresponding 3′ splice site as well as the score of each predicted branch point. The statistical significance of branch point score differences between differentially spliced cassette exons and native background cassette exons was assessed using Wilcoxon's rank-sum test.

*Anchoring sites upstream of branch points*. Sequences ranging from −25 nt to 4 nt relative to the predicted branch points were extracted, which includes the anchoring sites for the SF3B complex[21]. Observed frequencies of nucleotides at each specific position were visualized by WebLogo[85].

### Statistical analysis
To ensure rigor and transparency, the experimenters were unaware of mouse genotypes/grouping. Sample sizes and number of independent experiments were estimated by power analyses using an R script that takes the pre-specified effect size and type I and II errors as input arguments. Behavioral data scores were independently analyzed by two experimenters. For all the behavior and electrophysiology tests, we first analyzed sex-disaggregated data separately, and did not find any statistical significance between sexes. As such, data from both sexes were pooled for further analysis. Statistical analyses and graphing were performed using GraphPad Prism 10.0, Microsoft Excel, and

MATLAB and relevant bioinformatics tools and R packages. Figures were prepared using Adobe Creative Cloud. Shapiro–Wilk test and $F$ test were used to test normality and equal variance. Student $t$ test or two-way analysis of variations (with or without repeated measures, rmANOVA) were used when data passed normality and equal variance tests. Sidak's multiple comparison (MCT) test was used for post-hoc comparison following two-way ANOVA. A nonparametric Mann-Whitney $U$ test was used for non-normally distributed/ordinal type data (e.g., number of platform crosses). $p < 0.05$ was considered statistically significant for all tests.

## Reporting summary

Further information on research design is available in the Nature Portfolio Reporting Summary linked to this article.

## Data availability

The data supporting the findings of this study are available from the corresponding authors upon request. RNA Sequencing data generated in this study have been deposited to the NCBI Gene Expression Omnibus (GEO) (http://www.ncbi.nlm.nih.gov/geo) under accession number GSE212192. The full list of mass spectrometry-identified proteins is provided in Supplementary Data 1. The mass spectrometry proteomics data generated in this study have been deposited to the ProteomeXchange Consortium with the dataset identifier PXD050058. Source data are provided with this paper.

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

## Acknowledgements

Research reported in this publication was supported by the National Institute of Health under award number R01GM133850 and R01NS132083 (Y.Y.); R56HG012310 and R01GM088342 (Y.X.); R01MH128192 and R21AG078700 (S.Q.). The content is solely the responsibility of the authors and does not necessarily represent the official views of the National Institutes of Health. D.B.-L. is supported by NSERC 2021-03435 and CIHR 185869. We thank Drs. Douglas Black and Sika Zheng for their discussion of this project. We also thank Erin Keebaugh for providing helpful comments on scientific writing.

## Author contributions

Y.Y., Y.X. and S.Q. conceived the project. L.S. performed the majority of the biochemical and cellular experiments with the help of Z.W., Y.Z., H.P., X.T., D.L. and T.A. X.M., Y.C. and J.W. performed the mouse behavior test and ex vivo neuronal assays. Y.W. and Y.X. analyzed the RNA-seq data. N.S. prepared RNA-seq library. S.H. and T.C. established Prmt9 knockout mESCs. L.H. and D.B. performed structural analysis. Y.Y., S.Q. and Y.X. together wrote the manuscript with inputs from other authors.

## Competing interests

Y.X. is a scientific cofounder of Panorama Medicine. All other authors declare that they have no competing interests.
