## [Peer Review File · Nature Communications]

Loss-of-Function Mutation in PRMT9 Causes Abnormal Synapse Development by Dysregulation of RNA Alternative SplicingREVIEWER COMMENTS

Reviewer #1 (Remarks to the Author):

Protein Arginine Methyltransferase 9 (PRMT9) is the last human PRMT identified. So far very few substrates are known for PRMT9. In previous studies, the authors identified the splicing factor SF3B2 as a PRMT9 substrate. In this study, the authors established a Prmt9 conditional knock-out mouse model and linked PRMT9 function to neuron development. Knockout of PRMT9 in excitatory neurons led to aberrant synapse development and impaired learning and memory. Knock-in of a Sf3b2 arginine methylation-deficient mutant (R491K) in mouse model phenocopied Prmt9 knock-out, confirming the functional significance of PRMT9-catalyzed SF3B2 methylation. They performed RNA-seq analysis and identified the pre-mRNA splicing events altered in Prmt9 knock-out. Detailed analysis revealed that PRMT9-mediated SF3B2 arginine methylation regulates SF3B2 interaction with the RNA anchoring site for 3' splice site selection. Thus, this study demonstrates PRMT9 function in splicing regulation by fine-modulating SF3B2 activity via arginine methylation and links this function to autosomal recessive intellectual disability (ARID) with catalytically inactive PRMT9 G189R mutation. The study is novel and well performed. My comments are minor.

Minor comments:

1. Fig. 1e shows that all cells in Flag-WT sample were stained with anti-SF3B2 R508me2S, but only less than half of cells were stained with anti-Flag. How can SF3B2 be methylated in PRMT9 KO cells without Flag-WT expression? Were all cells transfected?
2. Fig.5d, the authors did not indicate whether the enriched pathway of alternative splicing events is relative to all splicing events or all genes. If they are enriched in comparison to all genes, they should also determine whether all splicing events are also enriched in the same pathway.
3. Fig.1b, the vertical SF3B2 should not be labeled in parallel with PRMT9s as SF3B2 is not a form of PRMT9. Change it to "-" and extend the horizontal SF3B2 label to the last four lanes. Replace SF3B2* with SF3B2 a.a. 400-550 (write in two lines) instead of defining it in the figure legend.
4. Fig.2a, anti-Flag WB of Flag-IP shows no ubiquitination of F-WT and anti-ubiquitin WB show some ubiquitination, suggesting that the ubiquitinated proteins immunoprecipitated by anti-Flag in F-WT sample are not PRMT9 but PRMT9-associated proteins. Labelling Ubn but not PRMT9-ubn in the figure.
5. Perform Flag-PRMT9, Flag-PRMT9-G189R half-life assay with MG132 treatment.
6. Fig.2f, the anti-UBE3C/Flag label is not clear although I understand anti-UBE3C for left and anti-Flag for right panel. Label the two panels separately.
7. Are the higher bands in the anti-GST WB of GST-UBE3C WT samples auto-ubiquitinated GST-UBE3C?
8. Extended Data Fig.5e, specify what MAVS* means.

Reviewer #2 (Remarks to the Author):

This study reports the role of PRMT9 in the regulation of alternative splicing related to synaptic proteins and learning and memory behaviors. The authors also find that SF3B2 is a major downstream target of PRMT9. These conclusions across the PRMT9-SF3B2 axis are solidly supported by both in vitro and in vivo results.

Major comments:

1. Fig. 5. It is unclear if the cause of reduced LTP is due to a decrease in NMDAR function or impaired AMPAR levels/trafficking. This should be clarified by further discussion. In addition, is LTD also changed? I am asking this question because behavioral flexibility is known to be suppressed when LTD is decreased. In addition, decreases in both LTP and LTD can strongly support the idea that NMDAR function is suppressed in addition to AMPAR function.

2. Comparing the learning and behavioral phenotypes in two different mouse lines (Prmt9 and Sf3b2) is an impressive approach. The authors measure learning and memory using the Morris water maze test and Sf3b2 mice but do not measure LTP, a critical synaptic measure of learning and memory, in the hippocampus as in Prmt9 mice. This would further support the hypothesis that Prmt9 mainly acts on SF3B2 in vivo.

3. The authors demonstrate the enrichment of alternatively spliced transcripts in genes with synapse/NMDAR-related functions. It would be nice if the authors could perform SynGO analysis to determine the extent of synaptic enrichments and have a better idea on the functions of the synapse genes (i.e. pre vs. postsynaptic localization, and functions within subcompartments of synapses).

Minor comments:

1. The authors conclude that Prmt9 loss-of-function has a cell-autonomous role based on the reduced excitatory synapses in Fig. 4e,f. However, the data does seem to justify this conclusion in my opinion. This should be clarified.

2. Figure 3h-j. The mutant mice show reductions in fear memory acquisition and retrieval. Because the fear acquisition is reduced, it is hard to tell if the fear memory is also reduced. This should be clarified. In addition, it is unclear when the test-phase experiment was performed (24 hours after the acquisition?).

3. AMPA receptor contents in excitatory synapses and spine head size have been well correlated with each other in the literature. This should be discussed.

Reviewer #3 (Remarks to the Author):

In this manuscript, Shen and colleagues demonstrated how a specific arginine methyltransferase, PRMT9, selectively methylates the splicing factor SF3B2, thereby regulating its splicing activity by influencing 3' splice site selection. This work is a follow up study of their previous work (refs. 12, 13) which determined that PRMT9 methylates SF3B2. This study extended their previous work in many aspects using an array of molecular, cellular and in vivo approaches, including: 1) G189R mutation found in ARID resulted in ubiquitination by UBE3C and protein degradation. 2) SF3B2 is the major, if not only, PRMT9 substrate that is methylated at a single arginine site (R508) in both human and mouse cells, suggesting a conserved function of SF3B2 methylation in regulating RNA splicing. 3) based on the PRMT9 G189R loss-of-function model, the authors showed that PRMT9 depletion in excitatory neurons in mice (PRMT9 cKO) leads to aberrant synapse development and impaired learning and memory, thus linking PRMT9 loss of function with ARID. 4) By generating a complementary SF3B2 methylation deficient knock-in mouse model, the authors were able to recapitulate similar defects seen in the PRMT9 cKO mouse model, thus postulating SF3B2 methylation deficiency drives the PRMT9 cKO phenotype. 5) PRMT9 deficiency resulted in global splicing switches, as identified by RNA-seq. Evaluation on a select set of exons demonstrated that these splicing defects can be recapitulated by Sf3b2 R491k mutation with loss of methylation. 6) Mechanistically, SF3B2 R508 methylation can reduce/block the interaction of SF3B2 with the anchoring site, which is located ~13nt upstream of BPS, thus reducing the selection of the 3' splice site. Overall, this work represents a very well designed, thorough, and comprehensive analysis of splicing regulatory mechanisms of PRMT9/SF3B2, and their functional significance in neurodevelopment, which is a significant addition to the relevant literature.

I have only several minor points which can hopefully improve the rigor/clarity of the study.

1. Fig. 1c, given that the G189R-mutant is quite unstable and very dramatically lowly expressed

compared to WT (Extended data Fig. 2a)—how do the authors explain the similar levels of IP'ed αFlag for WT and G198R samples? Did the authors load more sample from the G198R lysate compared to WT to compensate for its instability? Same for Figure 1d, 2a,

2, In Fig. 3 and Extended Figure 4, could the authors elaborate why the number of mice is different in different behavioral assays. Of note, in the Morris Water Maze test the number of mice is different between the training/acquisition, probe, and reverse learning trial. In addition, the authors should provide the number of mice and their respective sex for each assay in a separate section in Methods for clarity.

3. In Fig.4e, the puncta density for PSD95+ does not seem to change significantly (at least by eye) between the WT and cKO mice. Could the authors elaborate how they calculate the puncta density and provide a more representative picture.

4. Could the authors explain why they chose to use hippocampus tissue from Prmt9 KO (and respective littermates) female mice for the RNA-seq analysis. Did they use female mice in all the behavioral tests as well since they do not specify the sex in the manuscript?

5. Fig. 6, overall, the difference between Prmt9 regulated exons and control exons is quite small. I wonder whether more clear difference can be observed in Prmt9 targets defined with more stringent criteria.

6. The nomenclature of "native" vs. "degenerate" anchoring sites can be confusing. Maybe consensus vs. nonconsensus?

7. The authors might want to clarify whether 13 nt between the R508 interacting nucleotides and the BPS is strictly required. The anchoring site motif highlighted in Fig. 6e has different distances from the BPS.

8. Is there any insight from the structure data why only the interaction between R508 and purine is affected by methylation?

9. Throughout the text, Degenerated -> degenerate

10. P10, "Using native total lysate from HeLa cells as methylation substrates, we found that recombinant PRMT9 only methylates a single band that corresponds to the SF3B2 protein in PRMT9 KO, but not WT lysates." I suggest rephrase "a single band" with "one predominant band".

11. "To define the role of PRMT9-regualted RNA splicing in synapse development and function". Typo in "regualted".

12. P11, "Note that we introduced two synonymous mutations to prevent re-cutting by CRISPR/Cas9 (Extended Data Fig. 7a)." This statement lacks necessary contexts and is confusing.

We would like to thank all the reviewers for their insightful comments. Below are detailed point-by-point responses to the comments with our responses in *blue* text and the original comments in *black*. The sections of the main text that have been modified are highlighted in *red*.

Detailed Responses to Reviewer Comments

Reviewer #1 (Remarks to the Author):

Protein Arginine Methyltransferase 9 (PRMT9) is the last human PRMT identified. So far very few substrates are known for PRMT9. In previous studies, the authors identified the splicing factor SF3B2 as a PRMT9 substrate. In this study, the authors established a Prmt9 conditional knock-out mouse model and linked PRMT9 function to neuron development. Knockout of PRMT9 in excitatory neurons led to aberrant synapse development and impaired learning and memory. Knock-in of a Sf3b2 arginine methylation-deficient mutant (R491K) in mouse model phenocopied Prmt9 knock-out, confirming the functional significance of PRMT9-catalyzed SF3B2 methylation. They performed RNA-seq analysis and identified the pre-mRNA splicing events altered in Prmt9 knock-out. Detailed analysis revealed that PRMT9-mediated SF3B2 arginine methylation regulates SF3B2 interaction with the RNA anchoring site for 3' splice site selection. Thus, this study demonstrates PRMT9 function in splicing regulation by fine-modulating SF3B2 activity via arginine methylation and links this function to autosomal recessive intellectual disability (ARID) with catalytically inactive PRMT9 G189R mutation. The study is novel and well performed. My comments are minor.

We thank the reviewer for the highly positive comments about our work, specifically that “the study is novel and well performed”. The reviewer brought up a few minor points for clarification to strengthen the manuscript. We have addressed these requests in the revised manuscript, as detailed below.

Minor comments:

1. Fig. 1e shows that all cells in Flag-WT sample were stained with anti-SF3B2 R508me2S, but only less than half of cells were stained with anti-Flag. How can SF3B2 be methylated in PRMT9 KO cells without Flag-WT expression? Were all cells transfected?

We agree with the reviewer that the expression level of Flag-PRMT9 WT does not seem to correlate with the signal intensity of SF3B2 R508me2s staining. We reason that at least 3 factors could potentially contribute to this observation: 1) majority of the cells are transfected with the Flag-WT plasmid, but its expression level varies in individual cells. When we increase the brightness of the image, more cells are seen as transfected (**Figure R1**). The polyjet transfection reagent that we used in this study has a reported 88% transfection efficiency in HeLa cells (<https://signagen.com/In-Vitro-DNA-Transfection-Reagents/SL100688/PolyJet-DNA-In-Vitro-Transfection-Reagent>); 2) It is possible that the expression of Flag-WT in some cells has peaked and started to decline by the time this image was taken, whereas the SF3B2 R508me2s signal remains stable. Such results also indicate that it is less likely that SF3B2 R508me2s undergoes active demethylation; and 3) It is also possible that a relatively small amount or low level of rescue expression of Flag-WT is sufficient to restore majority of SF3B2 R508me2s level, thus higher or overexpression of PRMT9 does not necessarily lead to further increase of SF3B2 R508 methylation. This phenomenon has been observed with CARM1 and its methylation substrate PABP1.

CARM1 methylates chromatin remodeling factor BAF155 to enhance tumor progression and metastasis. Wang L, Zhao Z, Meyer MB, Saha S, Yu M, Guo A, Wisinski KB, Huang W, Cai W, Pike JW, Yuan M, Ahlquist P, Xu W. *Cancer Cell*. 2014 Jan 13;25(1):21-36. PMID: 24434208

Note that by adjusting the brightness and contrast, we were able to see cells that are negative for SF3B2 R508me2s staining (**Figure R1**, white arrows) and these cells are also negative for Flag-WT. Flag-G189R does not restore any SF3B2 R508me2s signal.

Figure R1. Increasing the brightness of Fig. 1e reveals that cells positive for SF3B2 R508me2s staining exhibited variable levels of Flag-WT, whereas cells negative for SF3B2 R508me2s staining are negative for Flag-WT. Flag-G189R does not restore any SF3B2 R508me2s signal.

2. Fig. 5d, the authors did not indicate whether the enriched pathway of alternative splicing events is relative to all splicing events or all genes. If they are enriched in comparison to all genes, they should also determine whether all splicing events are also enriched in the same pathway.

We thank the reviewer for this question. To clarify this, for the pathway enrichment analysis, we analyzed the enrichment of pathways in genes with significant splicing alterations in the SE and A3SS categories (foreground gene list) against a customized background gene list, in which we excluded genes expressed at very low levels (DeSeq2 baseMean value < 5) to eliminate potential bias resulting from gene expression on differential splicing analysis. Please see detailed description in the methods section.

To address the reviewer's question and test if all alternative splicing events are also enriched in the same pathways, we used 8,036 genes that exhibit SE or A3SS from all alternatively spliced genes identified from the RNA-seq data (**Extended Data Fig. 6c**) as the foreground gene list, which was then compared against the same background gene list. As shown in **Figure R2**, although several brain/neuronal related pathways are also enriched, likely due to the prevalence of alternative splicing in neuronal expressed genes, these enriched pathways exhibited relatively lower odds ratio than in the PRMT9-regulated, differentially spliced genes. Furthermore, synapse-specific pathways, such as "Activation of NMDA receptors upon glutamate binding", "Unblocking of NMDA receptor, glutamate binding and activation" are significantly enriched in PRMT9-regulated alternative splicing events but not among the top 10 enriched pathways in all alternative

splicing events, suggesting that PRMT9-regulated alternative splicing is more related to synapse development and function.

Figure R2. Pathway enrichment analysis of alternatively spliced genes. Pathways were grouped by database resources (BioPlanet pathway or WikiPathway). Only the top 10 enriched pathways are shown for each pathway database. 8,036 genes with SE or A3SS events were used as the foreground gene list, and a customized gene list excluding lowly expressed genes were used as the background gene list. The length of bars depicts the Benjamini-Hochberg adjusted p values calculated from a hypergeometric test. Dashed line represents adjusted p value of 0.05. Odds ratio of the enrichment is indicated by bar opacity.

To further determine the extent of synaptic enrichments, we performed the SynGO analysis (<https://www.syngoportal.org/>) on the differentially spliced genes identified between WT and Prmt9 cKO. As shown in **Figure R3**, we observed a significant enrichment of PRMT9-regulated differentially spliced genes to be localized in postsynapse components. The enriched child terms include postsynapse, postsynaptic specialization, postsynaptic density, and postsynaptic density membrane. This analysis revealed a potentially more specialized function of PRMT9-regulated alternative splicing in postsynapse function. The new data was included as the **new Extended Data Fig. 6h**.

Figure R3. SynGO enrichment analysis of Prmt9-regulated alternative splicing events, specifically SE and A3SS. The "brain expressed" background gene set downloaded from the SynGO database was selected, containing 18,035 unique genes among which 1,225 overlap with SynGO annotated synapse genes. In total, 175 out of 1,418 genes with Prmt9-regulated SE/A3SS events were mapped to SynGO annotated synapse genes. Parental GO terms are shown in inner circles and their corresponding child terms are shown in outer circles. Top level GO terms and the enriched child terms ($-\log_{10} Q\text{-value} \geq 6$) are text-labeled.

3. Fig.1b, the vertical SF3B2 should not be labeled in parallel with PRMT9s as SF3B2 is not a form of PRMT9. Change it to “-“ and extend the horizontal SF3B2 label to the last four lanes. Replace SF3B2* with SF3B2 a.a. 400-550 (write in two lines) instead of defining it in the figure legend.

We thank the reviewer for this suggestion. The **new Fig. 1b** has been revised accordingly.

4. Fig.2a, anti-Flag WB of Flag-IP shows no ubiquitination of F-WT and anti-ubiquitin WB show some ubiquitination, suggesting that the ubiquitinated proteins immunoprecipitated by anti-Flag in F-WT sample are not PRMT9 but PRMT9-associated proteins. Labelling Ubn but not PRMT9-ubn in the figure.

We thank the reviewer for pointing this out. To clarify this, for PRMT9 ubiquitination detection, we performed immunoprecipitation using RIPA lysis buffer that contains 0.1% SDS, aiming to disrupt potential PRMT9 interacting proteins, which, otherwise, might contribute to the ubiquitination signal. We think the differences in anti-Flag vs. anti-Ub detection is likely due to different sensitivity/epitope characteristics of these two antibodies. However, we cannot exclude the possibility that strong PRMT9 interaction proteins might still be co-immunoprecipitated under this stringent condition, as the reviewer indicated. We agree that it is appropriate to label it as Ubn (**new Fig. 2a and Fig. 2b**).

5. Perform Flag-PRMT9, Flag-PRMT9-G189R half-life assay with MG132 treatment.

We thank the reviewer for this suggestion. We performed this experiment, and the new results were included as the **new Extended Data Fig. 2b**. Consistent with our conclusion that G189R is degraded through ubiquitin-proteasome pathway, MG132 treatment stabilizes G189R protein and extends its half-life.

6. Fig.2f, the anti-UBE3C/Flag label is not clear although I understand anti-UBE3C for left and anti-Flag for right panel. Label the two panels separately.

As the reviewer suggested, we now labeled the two panels separately (**new Fig. 2f**).

7. Are the higher bands in the anti-GST WB of GST-UBE3C WT samples auto-ubiquitinated GST-UBE3C?

We thank the reviewer for this question. It is likely that the higher bands in the anti-GST blot is due to UBE3C auto-ubiquitination, as it is missing in E3 ligase activity deficient mutant (C1051S) or in the absence of E1/E2. Notably, it has been reported that UBE3C is auto-ubiquitinated at lysine 903 (K903). Please see the reference below. Thus, we now indicated this band as “* UBE3C auto-ubiquitination” in **new Fig. 2i**.

Crystal structure of HECT domain of UBE3C E3 ligase and its ubiquitination activity.
Singh S, Sivaraman J. *Biochem J.* 2020 Mar 13;477(5):905-923. PMID: 32039437

8. Extended Data Fig.5e, specify what MAVS* means.

As the reviewer suggested, we now added the description for MAVS* in the figure legend for the **Extended Data Fig. 5e**.

Reviewer #2 (Remarks to the Author):

This study reports the role of PRMT9 in the regulation of alternative splicing related to synaptic proteins and learning and memory behaviors. The authors also find that SF3B2 is a major downstream target of PRMT9. These conclusions across the PRMT9-SF3B2 axis are solidly supported by both in vitro and in vivo results.

Major comments:

1. Fig. 5. It is unclear if the cause of reduced LTP is due to a decrease in NMDAR function or impaired AMPAR levels/trafficking. This should be clarified by further discussion. In addition, is LTD also changed? I am asking this question because behavioral flexibility is known to be suppressed when LTD is decreased. In addition, decreases in both LTP and LTD can strongly support the idea that NMDAR function is suppressed in addition to AMPAR function.

We thank the reviewer for these questions. We agree that impaired LTP could be due to impaired NMDAR function and/or AMPAR trafficking. It is known that LTP drives AMPAR insertion following patterned neural activity. In the initial submission, we only reported LTP results. As the reviewer suggested, we now have included LTD data (which were collected only in a subset of mice, n = 6 slices) as the **new Extended Data Fig. 4i and 4j**. Together, the LTP and LTD data strongly support a suppressed neuronal plasticity.

2. Comparing the learning and behavioral phenotypes in two different mouse lines (Prmt9 and Sf3b2) is an impressive approach. The authors measure learning and memory using the Morris water maze test and Sf3b2 mice but do not measure LTP, a critical synaptic measure of learning and memory, in the hippocampus as in Prmt9 mice. This would further support the hypothesis that Prmt9 mainly acts on SF3B2 in vivo.

We thank the reviewer for bringing up this important point. We now included LTP data for this mouse line as well. Sf3b2 R491K knock-in mice show similarly impaired LTP responses as that observed with Prmt9 cKO. Note that these data were collected in mice ~2 months of age. The new data was included as the **new Extended Data Fig. 7h and 7i**.

3. The authors demonstrate the enrichment of alternatively spliced transcripts in genes with synapse/NMDAR-related functions. It would be nice if the authors could perform SynGO analysis to determine the extent of synaptic enrichments and have a better idea on the functions of the synapse genes (i.e. pre vs. postsynaptic localization, and functions within subcompartments of synapses).

We thank the reviewer for this suggestion. Following this suggestion, we performed the SynGO analysis (<https://www.syngoportal.org/>) on the differentially spliced genes identified between WT and Prmt9 cKO. As shown in **Figure R3**, we observed a significant enrichment of PRMT9-regulated differentially spliced genes that are localized in postsynaptic components. The enriched child terms include postsynapse, postsynaptic specialization, postsynaptic density, and postsynaptic density membrane. This analysis revealed a potentially more specialized function of PRMT9-regulated alternative splicing in postsynapse function. The new data was included as the **new Extended Data Fig. 6h**.

Figure R3. SynGO enrichment analysis of Prmt9-regulated alternative splicing events, specifically SE and A3SS. The "brain expressed" background gene set downloaded from the SynGO database was selected, containing 18,035 unique genes among which 1,225 overlap with SynGO annotated synapse genes. In total, 175 out of 1,418 genes with Prmt9-regulated SE/A3SS events were mapped to SynGO annotated synapse genes. Parental GO terms are shown in inner circles and their corresponding child terms are shown in outer circles. Top level GO terms and the enriched child terms ($-\log_{10} Q\text{-value} \geq 6$) are text-labeled.

Minor comments:

1. The authors conclude that Prmt9 loss-of-function has a cell-autonomous role based on the reduced excitatory synapses in Fig. 4e, f. However, the data does seem to justify this conclusion in my opinion. This should be clarified.

We now modified our Results section to further clarify this point. The results were obtained in low density hippocampal neurons grown *in vitro* on coverslips, which is devoid of *in vivo* developing conditions with network activities and glial-neuron interactions. This is what we meant by stating 'cell autonomous'.

2. Figure 3h-j. The mutant mice show reductions in fear memory acquisition and retrieval. Because the fear acquisition is reduced, it is hard to tell if the fear memory is also reduced. This should be clarified. In addition, it is unclear when the test-phase experiment was performed (24 hours after the acquisition?).

We thank the reviewer for this question. We agree that Fig. 3h should be better clarified with more details. In our initial submission, we cited our previous publication (Xia *et al.*, 2021) for a more detailed fear conditioning testing protocol. Fig. 3h is for the fear conditioning learning through 5 sessions in one day. Fig. 3i is for the contextual recall, which is a different memory modality. As such we infer impaired Pavlovian learning capacity in Prmt9 cKO mice. The reviewer is correct that contextual recall was conducted at 24 h after fear learning acquisition. We have now further clarified this in the Results section.

3. AMPA receptor contents in excitatory synapses and spine head size have been well correlated with each other in the literature. This should be discussed.

We thank the reviewer for bringing up this great point. We have now modified our discussion to reflect this: "Specifically, alternative splicing can directly impact glutamate receptors, plasticity, and maturation of cortical circuits. It is known that spine size and density are highly correlated with glutamate receptor content and degree of maturation. Our observation that CA1 neurons from Prmt9 cKO mice show decreased spine density and sizes, and that cultured primary cKO neurons exhibit reduced numbers of putative functional synapses further support a critical role of PRMT9/SF3B2-mediated splicing in neurodevelopment."

Reviewer #3 (Remarks to the Author):

In this manuscript, Shen and colleagues demonstrated how a specific arginine methyltransferase, PRMT9, selectively methylates the splicing factor SF3B2, thereby regulating its splicing activity by influencing 3' splice site selection. This work is a follow up study of their previous work (refs. 12, 13) which determined that PRMT9 methylates SF3B2. This study extended their previous work in many aspects using an array of molecular, cellular and in vivo approaches, including: 1) G189R mutation found in ARID resulted in ubiquitination by UBE3C and protein degradation. 2) SF3B2 is the major, if not only, PRMT9 substrate that is methylated at a single arginine site (R508) in both human and mouse cells, suggesting a conserved function of SF3B2 methylation in regulating RNA splicing. 3) based on the PRMT9 G189R loss-of-function model, the authors showed that PRMT9 depletion in excitatory neurons in mice (PRMT9 cKO) leads to aberrant synapse development and impaired learning and memory, thus linking PRMT9 loss of function with ARID. 4) By generating a complementary SF3B2 methylation deficient knock-in mouse model, the authors were able to recapitulate similar defects seen in the PRMT9 cKO mouse model, thus postulating SF3B2 methylation deficiency drives the PRMT9 cKO phenotype. 5) PRMT9 deficiency resulted in global splicing switches, as identified by RNA-seq. Evaluation on a select set of exons demonstrated that these splicing defects can be recapitulated by Sf3b2 R491k mutation with loss of methylation. 6) Mechanistically, SF3B2 R508 methylation can reduce/block the interaction of SF3B2 with the anchoring site, which is located ~13nt upstream of BPS, thus reducing the selection of the 3' splice site. Overall, this work represents a very well designed, thorough, and comprehensive analysis of splicing regulatory mechanisms of PRMT9/SF3B2, and their functional significance in neurodevelopment, which is a significant addition to the relevant literature.

I have only several minor points which can hopefully improve the rigor/clarity of the study.

1. Fig. 1c, given that the G189R-mutant is quite unstable and very dramatically lowly expressed compared to WT (Extended data Fig. 2a)—how do the authors explain the similar levels of IP'ed α Flag for WT and G189R samples? Did the authors load more sample from the G189R lysate compared to WT to compensate for its instability? Same for Figure 1d, 2a,

We thank the reviewer for raising this technical question. To compensate the low expression level of G189R protein (due to ubiquitination and degradation), we transfected different amount of WT and G189R plasmids (often in a ratio of 1: 5-8) to ensure cells express similar levels of both proteins to begin with. The loading volume for both input and IP samples were always kept the same.

2. In Fig. 3 and Extended Figure 4, could the authors elaborate why the number of mice is different in different behavioral assays. Of note, in the Morris Water Maze test the number of mice is different between the training/acquisition, probe, and reverse learning trial. In addition, the authors should provide the number of mice and their respective sex for each assay in a separate section in Methods for clarity.

We thank the reviewer for this question. In the initial submission, we included all the mice that passed the performance criteria in each given training/testing session. For instance, there were a few mice that did not pass the criteria (e.g. immobility time for longer than 10 sec) during day1-8 training sessions, but met the criteria in the probe trial. As such probe trial had more numbers. We agree with the reviewer that it is less confusing to keep the numbers consistent throughout different tests. In this revised version, we present data from 6 mice for both WT (4M2F) and cKO

(3M3F) mice that met all the selection criteria through the acquisition/probe trial/reverse learning. We have included this information in Methods and also figure legends.

3. In Fig.4e, the puncta density for PSD95+ does not seem to change significantly (at least by eye) between the WT and cKO mice. Could the authors elaborate how they calculate the puncta density and provide a more representative picture.

We thank the reviewer for this suggestion. We now included a more detailed description on quantification of overlapping, co-localized puncta in Methods. These quantifications were done using Imaris. We first thresholded the red and green channels to calculate puncta density, then used the red channel to create a mask and overlay on the thresholded green channels to detect colocalized puncta size and the proportion of colocalization.

After examining the initial quantification of Fig. 4e, the original figure is still the most representative of pooled data. However, we agree with the reviewer that the original dendritic segment may not look very different by eye. We selected another segment from the same neuron that shows better distinction (and closer to the quantitative results) between the two genotypes. **Fig. 4e** is revised.

4. Could the authors explain why they chose to use hippocampus tissue from Prmt9 KO (and respective littermates) female mice for the RNA-seq analysis. Did they use female mice in all the behavioral tests as well since they do not specify the sex in the manuscript?

We thank the reviewer for this question. Our behavior and electrophysiology data were obtained from mice of mixed gender, and we did not observe any gender preferences in terms of the impaired learning and memory in Prmt9 cKO mice. To minimize the variations for the identification of RNA alternative splicing, we decided to use age- and gender- matched mice to perform the RNA-seq experiment. There is no particular reason for using the female mice. We acknowledge that some of the alternative splicing targets should be verified in male mice in future studies. The reason we chose the hippocampus, not cortical tissue, is that most of the behavior (learning and memory) and ephys plasticity (LTP and LTD) data are more relevant to the hippocampus function.

5. Fig. 6, overall, the difference between Prmt9 regulated exons and control exons is quite small. I wonder whether more clear difference can be observed in Prmt9 targets defined with more stringent criteria.

We thank the reviewer for this question. We performed additional analyses using a more stringent criterion to define Prmt9-regulated alternative splicing events, specifically a larger PSI value difference of over 0.1 or 0.15, as compared to the standard PSI value difference of over 0.05 (currently used). However, this did not make the difference in splice site scores or branch point scores between Prmt9-regulated exons and control exons larger, and the statistical significance often dropped likely due to a smaller number of events meeting the more stringent criterion. We note that the small difference observed is not unexpected and in fact mirrors similar observations made in other studies of splice site scores (e.g. PMID: 32813009), given that all splice sites investigated are still canonical splice sites and possess splice site characteristics.

One limitation of this study is that RNA-seq was performed in bulk hippocampus tissues, and the heterogenous cellular population could contribute to the mild splicing changes observed in Prmt9 cKO hippocampus. As an ongoing project, we plan to perform nanopore long-read single-cell

RNA-seq (scRNAseq) to identify cell-type specific gene expression and splicing changes resulting from Prmt9 cKO.

6. The nomenclature of “native” vs. “degenerate” anchoring sites can be confusing. Maybe consensus vs. nonconsensus?

We thank the reviewer for this suggestion, and we made the changes as recommended.

7. The authors might want to clarify whether 13 nt between the R508 interacting nucleotides and the BPS is strictly required. The anchoring site motif highlighted in Fig. 6e has different distances from the BPS.

8. Is there any insight from the structure data why only the interaction between R508 and purine is affected by methylation?

We thank the reviewer for these questions. We agree that although the anchoring site motif shows sequence variations in PRMT9-regulated alternative splicing when compared to the unaffected native sequences (**Fig. 6e**), their exact distance to the BPS is not definitive. We reason that this is likely due to the low-resolution of RNA-seq analysis in defining such event. The available Cryo-EM spliceosome structures all indicate a close proximity of R508 with 13nt. We are in the process of performing Cross-linking and immunoprecipitation (CLIP) followed by next-generation sequencing (CLIP-seq) to achieve single-nucleotide resolution mapping of SF3B2 binding sequence in WT, Prmt9 KO, and Sf3b2 methylation deficient (R491K) mouse hippocampus. We expect that results from these experiments will provide more clear sequence/motif information about PRMT9-regulated alternative splicing.

The reviewer raised a few very important questions that are the focus of our future research. From structure analysis, we were able to formulate several hypotheses to address these questions. **First, why -13nt seems more important than -12nt or -14 nt (Fig. 7c)?** During branch point recognition and 3' splice site selection, U2 snRNA engages with intron sequence and forms

Figure R4. Protein-RNA and RNA-RNA interactions in U2 snRNP remodeling and U2/intron duplex formation. Cryo-EM structures of various human spliceosome complex, including (A) pre-B complex (PDB: 6QX9), (B) B-act complex (PDB: 5Z56), and (C) 17S U2 snRNP (PDB: 7Q3L) were shown with a focused view at the SF3A3 – U2/Intron interface. The pi-stacking interaction of SF3A3 W395 with either -13nt (when it is G in AdML pre-mRNA) or -12nt (when -13nt is U in MINX-GG pre-mRNA) was highlighted. Note that when W395 forms Pi-stacking with -12nt G, the base of the -13nt U is flipped out.

U2/Intron duplex. The -13nt is located at the last potential nucleotide for the formation of U2/Intron duplex (**Figure R4A and 4B**). This position corresponds to the G25 in the branchpoint-interacting-stem-loop (BSL) structure formed in the 17S U2 snRNP (**Figure R4C**). Based on these structures,

it is likely that the Tryptophan 395 (W395) of SF3A3 plays a critical role in preventing the further extension of both the BSL and the U2/Intron duplex. **Second, why purines vs. pyrimidines at -13nt exhibited different splicing patterns (Fig. 7c)?** We reason that this is because of the potential interaction of -13nt with W395 of SF3A3. As seen in 17S U2 snRNP (7Q3L) and in pre-B complex (6QX9), W395 forms stable pi-stacking interaction with G25 of U2 snRNA in 17S, and with -13nt purine (G) in pre-B complex (AdML pre-mRNA). However, in 5Z56, where another pre-mRNA MINX-GG was used, W395 interacts with -12nt purine (G) and the -13nt pyrimidine (U) was flipped out towards SF3B2 R508. Because two-ring purines (A/G) are more advantageous for stable pi-stacking than the one-ring pyrimidines (C/U), we hypothesize that W395 would preferably interact with -13nt nucleotide (if it is A/G) or -12nt A/G nucleotide (if -13nt is C/U). The consequence of this differential interaction (W395: -13nt vs. W395: -12nt) could affect the stability of the assembled spliceosome, causing different splicing outcomes. **Last, why only purines at -13nt are affected by the loss of SF3B2 R508 methylation (Fig. 7d)?** As shown in the in vitro EMSA assay, hypomethylation of R508 enhances SF3B2 interaction with RNA (**Extended Data Fig. 8d**). This interaction could potentially interfere the pi-stacking interaction of SF3A3 W395 with purines at -13nt. Whereas if -13nt is a pyrimidine, its interaction with SF3A3 W395 might have already weakened, thus are less likely to be affected by the enhanced SF3B2-RNA interaction.

While actively testing these hypotheses, we acknowledge that this model could be over simplified, as it does not consider the dynamic impacts of these interactions on the structure/stability of U2 snRNA. The working hypothesis will be adjusted as more experimental evidence emerges.

9. Throughout the text, Degenerated -> degenerate

We thank the reviewer for this suggestion. We have made the changes as the reviewer suggested.

10. P10, "Using native total lysate from HeLa cells as methylation substrates, we found that recombinant PRMT9 only methylates a single band that corresponds to the SF3B2 protein in PRMT9 KO, but not WT lysates." I suggest rephrase "a single band" with "one predominant band".

We thank the reviewer for this suggestion. We rephrased the text as the reviewer suggested.

11. "To define the role of PRMT9-regulated RNA splicing in synapse development and function". Typo in "regulated".

We thank the reviewer for finding this typo. It is now corrected.

12. P11, "Note that we introduced two synonymous mutations to prevent re-cutting by CRISPR/Cas9 (Extended Data Fig. 7a)." This statement lacks necessary contexts and is confusing.

We thank the reviewer for pointing this out. To avoid confusion, we removed this sentence in the revised text, since the detailed information about the generation of R491K knock-in mice was included in the Methods section. See "Two synonymous mutations p.C488= (TGT to TGC) and p.K490= (AAG to AAA) were introduced to prevent the binding and re-cutting of the sequence by gRNA after homology-directed repair."

REVIEWERS' COMMENTS

Reviewer #1 (Remarks to the Author):

The authors have addressed my comments satisfactorily.

Reviewer #2 (Remarks to the Author):

The authors have fully addressed my review comments by performing additional LTP/LTD experiments and SynGO analyses. I do not have additional comments.

Reviewer #3 (Remarks to the Author):

The authors have addressed my previous comments. I have no additional comments on the revised manuscript.